# Dissecting infant leukemia developmental origins with a hemogenic gastruloid model

Denise Ragusa[1†], Chun Wai Suen[2†], Gabriel Torregrosa Cortes[3†], Fabio Pastorino[4†], Ayona Johns[1], Ylenia Cicirò[1], Liza Dijkhuis[1‡, §], Susanne van den Brink[5,6], Michele Cilli[7], Connor Byrne[1#], Giulia-Andreea Ionescu[1¶], Joana Cerveira[8], Kamil R Kranc[9,10], Victor Hernandez-Hernandez[1], Mirco Ponzoni[4], Anna Bigas[5,6], Jordi Garcia-Ojalvo[3], Alfonso Martínez Arias[3], Cristina Pina[1,11,12*]

*For correspondence: c.pina@sanquin.nl

†These authors contributed equally to this work

Present address: ‡Sanquin Research, Landsteiner Laboratory, Amsterdam, Netherlands; §University Medical Center, Amsterdam, Netherlands; #Faculty of Biology, Medicine and Health, University of Manchester, Manchester, United Kingdom; ¶Division of Infection and Immunity, Cardiff University School of Medicine, Cardiff, United Kingdom

[1]College of Health, Medicine and Life Sciences, Centre for Genome Engineering and Maintenance, Brunel University, London, United Kingdom; [2]Department of Genetics, University of Cambridge, Cambridge, United Kingdom; [3]Department of Medicine and Life Sciences, Universitat Pompeu Fabra, Barcelona, Spain; [4]Laboratory of Experimental Therapies in Oncology, IRCCS Istituto Giannina Gaslini, Genova, Italy; [5]Program in Cancer Research, Hospital del Mar Research Institute, CIBERONC, Barcelona, Spain; [6]Josep Carreras Leukemia Research Institute, Barcelona, Spain; [7]Animal Facility, IRCCS Policlinico San Martino, Genova, Italy; [8]Department of Pathology, University of Cambridge, Cambridge, United Kingdom; [9]Centre for Haemato-Oncology, Barts Cancer Institute, Queen Mary University of London, London, United Kingdom; [10]Centre for In Vivo Modelling, Institute of Cancer Research, Sutton, United Kingdom; [11]Sanquin Research, Landsteiner Laboratory, Amsterdam, Netherlands; [12]Amsterdam UMC Location Vrije Universiteit Amsterdam, Amsterdam, Netherlands

## eLife Assessment

This **valuable** study presents a mouse gastruloid system to generate successive waves of hematopoietic progenitors that in vivo would emerge during embryonic development. Although this newly revised manuscript has addressed some of the concerns raised during the first round of review, the study is still considered **incomplete**, as the claims are only partially supported. In particular, the claim of definitive wave hematopoietic progenitors being produced in the gastruloids, and their engraftment after transplantation, would benefit from further validation.

**Abstract** Current in vitro models of developmental blood formation lack spatio-temporal accuracy and weakly replicate successive waves of hematopoiesis. Herein, we describe a mouse embryonic stem cell (SC)-derived 3D hemogenic gastruloid (haemGx) that captures multi-wave blood formation, progenitor specification from hemogenic endothelium (HE), and generates hematopoietic progenitors capable of short-term engraftment of immunodeficient mice upon maturation in an in vivo niche. We took advantage of the haemGx model to interrogate the origins of infant acute myeloid leukemia (infAML). We focused on MNX1-driven leukemia, representing the commonest genetic abnormality unique to the infant group. Enforced MNX1 expression in haemGx promotes the expansion and in vitro transformation of yolk sac-like erythroid-myeloid progenitors at the HE-to-hematopoietic transition to faithfully recapitulate patient transcriptional signatures. By combining

phenotypic, functional, and transcriptional profiling, including at the single-cell level, we establish the haemGx as a useful new model for the study of normal and leukemic embryonic hematopoiesis.

## Introduction

Blood formation in the embryo is a multi-stage process that develops across multiple cellular niches, reflecting a complex hierarchy of tissue interactions. It involves sequential production of distinct cell types in so-called hematopoietic waves, separated in time and space (*Lacaud and Kouskoff, 2017*; *Costa et al., 2012*; *Medvinsky et al., 2011*; *Dzierzak and Bigas, 2018*). The first wave of hematopoiesis produces unipotent red blood cell and macrophage precursors in the yolk sac (YS) from an angioblast precursor which can also form endothelium; it occurs at mouse embryo day (E)7-E7.5 (*Fujimoto et al., 2001*; *McGrath and Palis, 2005*; *Moore and Metcalf, 1970*). The subsequent wave, known as definitive, produces: first, oligopotent erythro-myeloid progenitors (EMPs) in the YS (E8-E8.5); and later myelo-lymphoid progenitors (MLPs – E9.5-E10), multipotent progenitors (MPPs – E10-E11.5), and hematopoietic stem cells (HSCs – E10.5-E11.5), in the aorta-gonad-mesonephros (AGM) region of the embryo proper (*Ivanovs et al., 2011*; *McGrath et al., 2015*; *Palis et al., 1999*; *Medvinsky and Dzierzak, 1996*; *Medvinsky et al., 2011*). Definitive blood waves are characteristically specified from a specialized HE through a process of endothelial-to-hematopoietic transition (EHT) (*Zovein et al., 2008*; *Li et al., 2005*). HSC specification from HE is the last event in hematopoietic development and occurs in characteristic cell clusters found on the ventral wall of the dorsal aorta (*Ivanovs et al., 2011*; *Boisset et al., 2010*). All blood cell types enter circulation from their original locations and migrate to the fetal liver (FL) (*Ciriza et al., 2013*; *Ghiaur et al., 2008*). Most embryonic blood progenitors are transient and sustain blood production exclusively during embryonic development, through differentiation. In contrast, HSCs make little or no contribution to embryonic and fetal blood; instead, they proliferate (i.e. self-renew) to expand their numbers. At least a subset of HSCs migrates to the bone marrow (BM) niche at the end of gestation and balances self-renewal and differentiation to maintain blood production throughout the entire post-natal life (*Bowie et al., 2006*; *Kikuchi and Kondo, 2006*; *Ganuza et al., 2022*; *Yokomizo et al., 2022*).

Blood progenitors and HSCs can be targeted by genetic alterations that confer clonal advantage and/or drive initiation and development of leukemia. Leukemia is the most common malignancy and the most common cause of cancer-related death in the pediatric age group (*Steliarova-Foucher et al., 2017*; *Dong et al., 2020*). Several pediatric leukemias originate in utero, and a subset of these are driven by mutations exclusive to the pediatric group (*Masetti et al., 2015*; *Cazzola et al., 2020*). Such pediatric-exclusive mutations target blood cell types restricted to development or depend on signals that are specific to embryonic hematopoietic niches, effectively defining distinct biological entities to adult malignancies (*Cazzola et al., 2020*; *Bolouri et al., 2018*; *Chaudhury et al., 2018*). Dissection of these forms of developmental leukemia is confounded by the inability to robustly identify and isolate their embryonic cells of origin, and/or to fully recapitulate the disease in post-natal cells, which leads to failure in generating relevant targeted therapies (*Alexander and Mulligan, 2021*). One such example is t(7;12) (q36;p13) acute myeloid leukemia (AML) driven by rearrangement and overexpression of the *MNX1* gene locus: *MNX1*-rearranged (*MNX1-r*) AML is restricted to infants under the age of 2 and carries a high risk of relapse and poor survival (*Ragusa et al., 2023*). *MNX1* is a developmental gene required for specification of motor neurons and pancreatic acinar cells (*Thaler et al., 1999*; *Harrison et al., 1999*). MNX1 is not known to participate in embryonic hematopoiesis; however, it has been discordantly described to be expressed in some subsets of adult hematopoietic cells (*Deguchi and Kehrl, 1991*; *Nagel et al., 2005*; *Wildenhain et al., 2012*; *Taketani et al., 2008*). It is ectopically expressed in AML cells carrying one of a variable set of t(7;12) translocations which place the normally silent *MNX1* gene under the control of *ETV6* regulatory elements (*Weichenhan et al., 2023*; *Bousquets-Muñoz et al., 2024*; *Ragusa et al., 2023*; *Ballabio et al., 2009*). *ETV6* is required for hematopoietic cell specification from HE via VEGF signalling (*Ciau-Uitz et al., 2010*), potentially positioning *MNX1* within a critical embryonic hemogenic regulatory network.

Faithful in vitro recapitulation of developmental blood cell types and their embryonic niches has the potential to dissect the cellular and molecular mechanisms presiding at initiation of developmental leukemia, reveal prognostic biomarkers, and indicate therapeutic vulnerabilities which are otherwise challenging to test in the embryo. Despite widespread use of embryonic stem (ES) cell and induced

pluripotent stem (iPS) cell-based models of blood specification, a system which recapitulates the spatial and temporal complexity of embryonic blood formation within the respective time-dependent niches is still lacking (*Dijkhuis et al., 2023*). In the last few years, gastruloid models have emerged as robust avatars of early development, self-organizing processes of symmetry breaking, elongation, multi-axis formation, somitogenesis and early organogenesis, with remarkable similarity to embryo processes in space and time (*van den Brink and van Oudenaarden, 2021*; *Beccari et al., 2018*; *van den Brink et al., 2020*). Gastruloids are grown individually in a multi-well format that facilitates tracking and screening of developmental processes. Relevant to blood formation, a variant gastruloid culture system, which successfully organizes a heart primordium (*Rossi et al., 2021*), was also able to recapitulate YS blood formation with erythroid and myeloid cellularity (*Rossi et al., 2021*; *Rossi et al., 2022*).

Herein, we describe a new protocol of gastruloid formation, i.e., hemogenic gastruloids (haemGx), that captures putative YS-like and AGM-like waves of definitive blood progenitor specification from HE. The earlier wave of putative YS-like hematopoiesis includes EMP-like cells which further mature in the gastruloid culture. HaemGx AGM-like hematopoiesis captures the emergence of intra-aortic cluster-like cells, including cells capable of short-term multi-lineage hematopoietic engraftment of the spleen and BM of immunodeficient mice. We harnessed the embryonic hematopoietic context provided by haemGx to attest its utility in modelling leukemias with developmental origins, specifically by recapitulating *MNX1-r* infant leukemia by proxy of *MNX1* overexpression. MNX1-overexpressing haemGx showed expansion of cells at the HE-to-EMP transition, which became susceptible to transformation in vitro and recapitulated *MNX1-r* AML patient signatures, placing *MNX1*'s leukemogenic effects at a very early stage of developmental blood formation.

## Results
### HaemGxs capture time-dependent emergence of endothelium and hematopoietic cells

With the aim of recapitulating embryonic hematopoiesis with spatial and temporal accuracy, we modified existing gastruloid formation protocols to (1) promote the specification of hemogenic mesoderm derivatives, and (2) extend developmental time beyond the E8.0-equivalent of early developmental gastruloids (*van den Brink et al., 2014*) and the E9.0 timepoint of cardiogenic gastruloids (*Rossi et al., 2022*). We established a 216 hr haemGx protocol (*Figure 1A*) using a *Kdr(Flk1)-GFP* mouse ES cell line (*Jakobsson et al., 2010*), which allows the tracking of early hemato-endothelial specification by expression of the *Flk1* (i.e. *Kdr*) locus. The haemGx protocol includes an indispensable pulse of Activin A to induce hemato-endothelial programmes via TGF-β signalling (visible by proxy of GFP-tagged *Flk1* activation), in addition to the characteristic gastruloid patterning via WNT activation by CHI99021 supplementation at 48 hr (*Figure 1—figure supplement 1A and B*). We promoted hemato-endothelial programs through addition of VEGF and FGF2 at 72 hr (*Sroczynska et al., 2009*; *Figure 1A*). We observed extension of a polarized *Flk1-GFP*-expressing branched endothelial network (*Figure 1B*). Flow cytometry analysis revealed the onset of VE-cadherin$^+$ C-Kit$^+$ cells suggestive of EHT, with a peak at 120–144 hr (*Figure 1C–D*). This was followed by a surge in CD41$^+$ candidate hematopoietic cells at 144 hr (*Figure 1C–D*). In our experience, observation of early polarized activation of the *Flk1 locus* at 96 hr (*Figure 1B*) was predictive of CD41$^+$ detection 2 days later, suggesting that TGF-β activation in early patterning is required for later hematopoietic development, putatively by modifying mesoderm specification. We incorporated a 24 hr pulse of sonic hedgehog (Shh) between 144–168 hr to mimic the aortic patterning that precedes cluster formation (*Figure 1A*; *Rybtsov et al., 2014*). Continued exposure to VEGF after the Shh pulse resulted in specification of CD45$^+$ hematopoietic cells progenitors (*Figure 1C–E*), which were apparent after 168 hr. The proportion of CD45$^+$ cells was enhanced by the addition of SCF, FLT3L, and TPO (*Figure 1—figure supplement 1C and D*), a combination of cytokines routinely used for HSC maintenance; conversely, it did not require continued addition of FGF2 after 168 hr (*Figure 1—figure supplement 1E*), which was omitted. CD45$^+$ cells, some of which co-expressed C-Kit, were present in a small number of 'cluster'-like structures adjacent to Flk1-GFP$^+$ Kit$^+$ endothelium (*Figure 1E*). The process is overall reminiscent of hematopoietic progenitor cell budding from the aortic HE, suggesting recapitulation of hemogenic tissue architecture. Time-dependent specification of C-Kit$^+$, CD41$^+$, and CD45$^+$ cells in haemGx could be

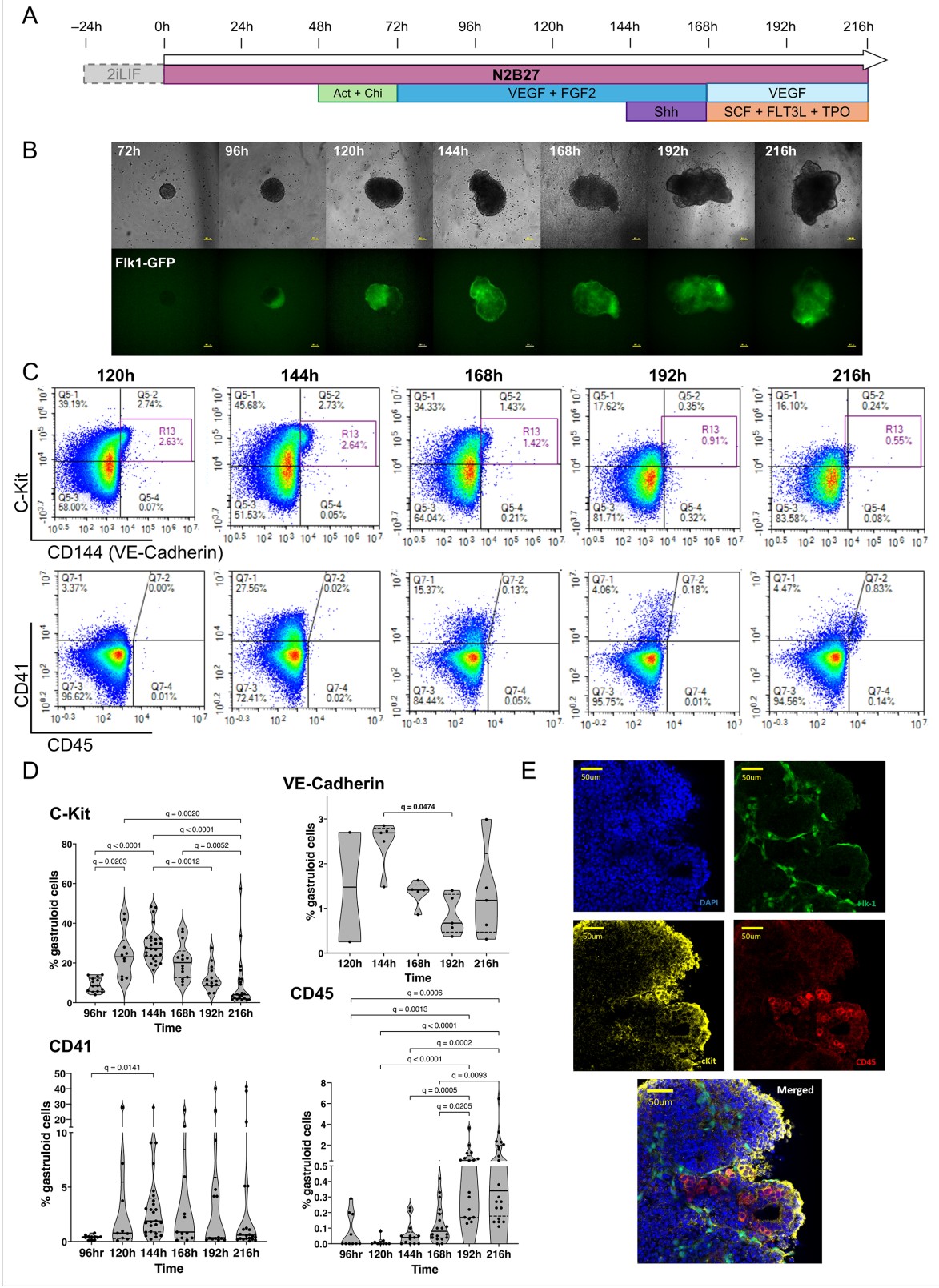

**Figure 1.** Hemogenic gastruloids (haemGx) produced from mES cells promote hemato-endothelial specification with spatio-temporally accurate ontogeny. (**A**) Timeline of mES cells assembly and culture into gastruloids over a 216 hr period with the addition of specified factors for the promotion of hemato-endothelial specification. The 24 hr pre-treatment in 2i+LIF was omitted when mES cultures showed no signs of differentiation. (**B**) Imaging of haemGx over time from 72 to 216 hr at 10 x magnification, showing the assembly and growth of the 3D structures and the polarization of the Flk-1-GFP

*Figure 1 continued on next page*

Figure 1 continued

marker from 96 hr; scale bar: 100 mm. (**C**) Flow cytometry timecourse analysis of haemGx for the presence of CD144 (VE-cadherin), C-Kit, CD41, and CD45 markers; representative plots. (**D**) Quantification of analysis in (**C**). Violin plots with median and interquartile range of up to 26 replicates; Kruskal-Wallis CD144 (p=0.0755), C-Kit (p<0.0001), CD41 (p<0.0342), CD45 (p<0.0001). Shown are significant Dunn's multiple comparisons tests; adjusted q-value. (**E**) Immunostaining of whole individual haemGx at 216 hr showing the localized expression of Flk1-GFP (green), C-Kit (yellow), and CD45 (red) markers; nuclear staining is in DAPI (blue); scale bar: 100 μm.

The online version of this article includes the following figure supplement(s) for figure 1:

**Figure supplement 1.** Optimization and reproducibility of the hemogenic gastruloid (haemGx) protocol.

obtained with other mouse ES cell lines, including commonly used E14Tg2a (E14) cells (**Figure 1—figure supplement 1F-H**), with successful specification of comparable frequencies of CD45$^+$ progenitors (**Figure 1—figure supplement 1I**), attesting to the reproducibility of the protocol across cell lines of different genetic backgrounds.

Overall, our extended haemGx model could capture time-dependent specification of CD41$^+$ and CD45$^+$ hematopoietic cells, with recapitulation of the cluster-like spatial arrangement of CD45$^+$ cell emergence from HE, thus showing initial promise as a tool for the exploration of normal and perturbed developmental hematopoiesis.

## HaemGx specify cells with phenotypic and functional hematopoietic progenitor characteristics

We confirmed the induction of endothelial patterning from the Flk1-GFP population by immuno-fluorescence staining showing the branched vascular-like network of Flk1 co-expressing CD31 (**Figure 2A**). Endothelial identity of Flk1-GFP$^+$ populations was also detectable by flow cytometry through the presence of endothelial markers CD31 and CD34 (**Figure 2—figure supplement 1A and B**). Given the co-expression of CD31 and CD34 with hematopoietic progenitors emerging from HE, we inspected CD41$^+$ and CD45$^+$ cells for the presence of these markers. Indeed, we found that subsets of CD41$^+$ and CD45$^+$ hematopoietic cells co-express CD31 and CD34 (**Figure 2B**, **Figure 2—figure supplement 1C and D**), with increased relative representation of hemato-endothelial markers within CD41$^+$ cells at 144 hr and CD45$^+$ at 216 hr (**Figure 2C**), consistent with two successive waves of hemogenic emergence from HE. CD45$^+$ cells from 216 hr haemGx also co-expressed C-Kit, Flk1-GFP, and VE-cadherin, a phenotype enriched in emergent hematopoietic stem and progenitor cells (HSPC) in the AGM (**Figure 2—figure supplement 1E and F**). The phenotype is detected in 1.8% of CD45$^+$ cells on average (**Figure 2—figure supplement 1G**), corresponding to an estimated 8 cells per haemGx. In agreement, analysis of the haemGx progenitor content at 216 hr using multipotential CFC assays (**Figure 2D–E**) revealed the presence of >50 CFC/haemGx, of which an average of seven have mixed-lineage granulocytic-monocytic-erythroid (GEM) potential, numerically matching the estimated number of HSPC. Other progenitors were predominantly granulo-monocytic (GM) and erythroid (E) progenitors (**Figure 2D–E**). Analysis of the cellularity of dissociated haemGx at the 216 hr timepoint using Giemsa-Wright stained cytospins revealed differentiated cells of G, M, E, and megakaryocytic lineages (**Figure 2—figure supplement 2A**), putatively reflecting the progeny of EMP-like cells.

We observed temporal progression of expression of key markers by real-time quantitative PCR (qPCR), configuring a sustained hemato-endothelial program from 120 hr through 216 hr (*Dll4* and *Mecom*), and the generation of hematopoietic populations based on the time-dependent patterns of *Myb*, *Runx1*, *Hlf*, and *Hoxa9* (**Figure 2F**). We note that *Myb* peaks at 144 hr, coincident with the emergence of CD41$^+$ cells, which we postulate to have YS-like EMP characteristics. On the other hand, *Runx1* has a bimodal pattern peaking at 144 hr, and later at 216 hr, suggesting the presence of two distinct hematopoietic waves. The later timepoint of 216 hr has significantly higher expression of both *Hoxa9* and *Hlf* raising the possibility that the second wave indeed corresponds to the specification of AGM-like HSPC. In order to further test the capture of YS-like and AGM-like hematopoietic waves, given the lack of markers that specifically associate with AGM hematopoiesis, we took advantage of the distinct regulatory requirements of polycomb-associated EZH2, which is necessary for EMP differentiation in the YS, but not for AGM-derived hematopoiesis or primitive erythroid progenitors (**Neo et al., 2018**; **Neo et al., 2021**). Upon EZH2 inhibition via treatment with GSK216 from 120 hr onwards, we observed a reduction of CD41$^+$ cells selectively at 144 hr, with no change in CD45$^+$ cells at the later timepoint of 216 hr (**Figure 2—figure supplement 2B and C**). This suggests that the cells

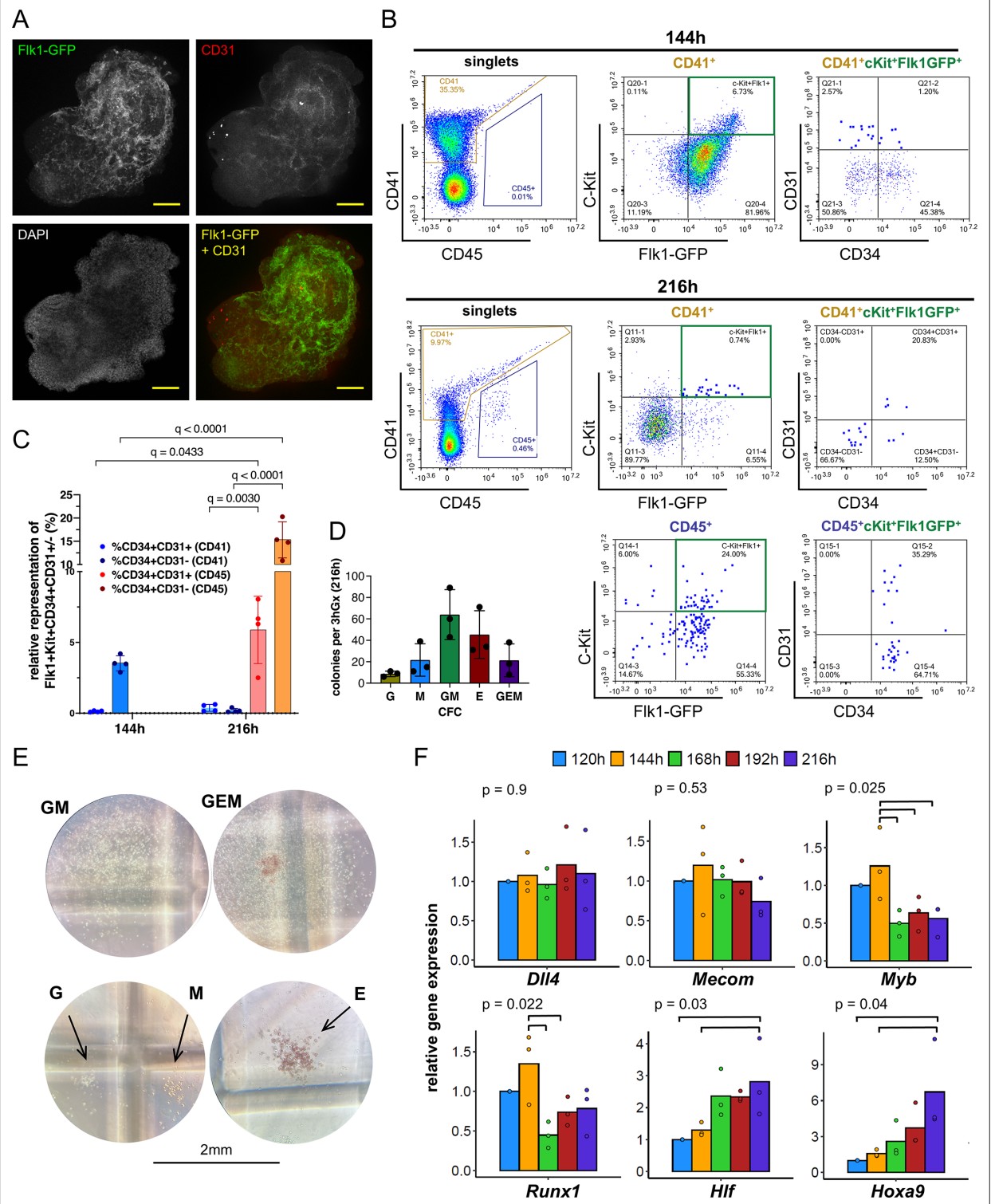

**Figure 2.** Hemogenic gastruloid (HaemGx) produce hematopoietic output following the emergence of specialized endothelium. (**A**) Immunostaining of whole individual haemGx at 144 hr showing the establishment of a vascular network by co-expression of CD31 and Flk1-GFP; nuclear staining is in DAPI (blue); scale bar: 100 μm. (**B**) Flow cytometry analysis of haemGx cells at 144 hr and 216 rhr stained for CD41, CD45, CD31, and CD34. (**C**) Quantification of flow cytometry analysis in (**B**) of CD41+ and CD45+ fractions co-expressing CD34 and CD31. Mixed effect analysis with a Šidàk test for multiple comparisons. (**D**) Colony-forming cell (CFC) assay of 216 hr-haemGx in multipotential methylcellulose-based medium. GEM: granulocyte-erythroid-monocyte; GM: granulocyte-monocyte; M: monocyte; E: erythroid. Please note that the medium does not include TPO and does not assess the presence of megakaryocytic progenitors. CFC frequency in 3 haemGx, n=3, mean ± SD. (**E**) Representative photographs of colonies quantified

*Figure 2 continued on next page*

*Figure 2 continued*

in C; scale bar = 2 mm. (**F**) Real-time quantitative (qPCR) analysis of expression of hemato-endothelial genes by timepoint. Relative gene expression fold change calculated by normalization to Ppia. Bars represent mean of three replicates and show individual data points; p-values by ANOVA across datapoints; post-hoc statistical significance between specific variables by Tukey's test and shown by brackets.

The online version of this article includes the following figure supplement(s) for figure 2:

**Figure supplement 1.** Multi-colour flow cytometry detection of surface markers in hemogenic gastruloid (haemGx).

**Figure supplement 2.** Characterization of hematopoietic output from hemogenic gastruloid (haemGx).

present at the later timepoint may not be the progeny of the former, thus supporting recapitulation of two independent hematopoietic waves, the latter putatively AGM. Additionally, the vulnerability of 144 hr CD41$^+$ to EZH2 inhibition suggests that they include YS-like EMP and do not predominantly reflect EZH2-independent primitive YS hematopoiesis.

## HaemGx progenitors capture time-dependent signatures of YS-like and AGM-like hematopoiesis

To better characterize the extent and progression of developmental hematopoiesis in haemGx, we performed single-cell RNA-sequencing (scRNA-seq) time-course analysis of haemGx cell specification. We sorted cells from two independent haemGx cultures at 120, 144, 168, 192, and 216 hr, and profiled a total of 846 cells using the Smart-Seq2 protocol (*Picelli et al., 2014*; *Figure 3—figure supplement 1A*). We analyzed unfractionated live single haemGx cells at 120–192 hr and enriched the dataset with sorted CD41$^+$ cells at their peak time of 144 hr, and sorted CD45$^+$ cells at 192 and 216 hr. At timepoints of significant enrichment in CD41$^+$ (144 hr) and CD45$^+$ cells (192 hr) (*Figure 1D*), we also sorted C-Kit$^+$ cells. Library preparation and sequencing generated an average of 120,000 reads/cell, which were mapped to an average of 4000 genes/cell, with minimal signs of cell stress/death as measured by mitochondrial DNA fraction (*Figure 3—figure supplement 1B*). Read and gene counts were similar between biological replicates, cell types, and at different timepoints (*Figure 3—figure supplement 1C*). The exception was 120 hr unfractionated haemGx cells, which despite similar sequencing depth (average read counts), were mapped to twice the number of genes (*Figure 3—figure supplement 1C*), possibly reflecting multi-gene program priming at the onset of hemogenic specification. We selected highly varying genes (HVG) before principal component analysis (PCA) dimensionality reduction and retained the most relevant dimensions. In the PCA reduced space, we constructed a KNN graph and used uniform manifold approximation and projection (UMAP) to visualize the data on two dimensions, looking for cell communities using Leiden clustering (*Figure 3—figure supplement 1D*). We identified 12 cell clusters of which 2 (clusters 0 and 5) almost exactly mapped to C-Kit$^+$ cells (also positive for Sca-1), 2 (clusters 1 and 8) contained CD45$^+$ sorted cells, and cluster 4 uniquely captured the CD41$^+$ cells observed at 144 hr (*Figure 3—figure supplement 1D*). Although some unfractionated cells overlapped with the hemogenic cell clusters (*Figure 3—figure supplement 1D*, middle panel), most occupied different transcriptional spaces, reflecting the relative frequency of hematopoietic cells and suggesting the presence of potential hemogenic niches.

To explore tissue and lineage affiliation of the different clusters, we performed differential gene expression against all other cells using the Wilcoxon ranking test, and established cluster classifier gene lists (*Supplementary file 1*). Top classifier genes for clusters 0 and 5 had endothelial affiliation (*Supplementary file 1* and *Supplementary file 2*), and the cellular composition of the clusters broadly reflected their time signature (*Figure 3—figure supplement 1D*): cluster 5 comprised 144 hr cells and cluster 0 included most C-Kit$^+$ cells at 192 hr. Cluster 5 occupied the vicinity of 144 hr CD41$^+$ cells (*Figure 3—figure supplement 1D*), which carried an erythroid-biased signature (*Supplementary file 2*). On the other hand, 192/216h-CD45$^+$ cells in cluster 8 expressed myeloid and lymphoid genes (*Supplementary file 1* and *Supplementary file 2*). The remaining clusters largely corresponded to time-restricted unfractionated cells (*Figure 3—figure supplement 1D*, bottom panel), with clusters of cells present at later timepoints matching mesenchymal stromal cell and autonomic neuron signatures (*Figure 3—figure supplement 1E*), potentially configuring incipient formation of niches relevant to production of HSCs (*Kapeni et al., 2022*; *Fitch et al., 2012*).

To specifically explore the extent to which haemGx capture putative YS and AGM-like hematopoiesis, we re-clustered cells sorted on hemato-endothelial markers CD41, C-Kit, and CD45 at the

144 hr, 192 hr, and 216 hr timepoints (*Figure 3A*). We mapped the expression pattern of endothelial (*Pecam* and *Cd34*), HE (*Mecom* and *Cd44*) and hematopoietic (*Runx1*, *Mllt3*, *Hoxa9*, and *Hlf*) markers across the time-dependent clusters (*Figure 3B*). We observed a coincidence of *Pecam*, *Cd34*, and *Mecom* expression with clusters 0 and 3, which capture C-Kit+ cells at 144 and 192 hr, respectively. Some positive cells were also seen within CD41+ and CD45+-enriched clusters 2, and 1 and 4, respectively, supporting the presence of phenotypic progenitors, as suggested by flow cytometry analysis (see *Figure 2*). CD41+ and CD45+-sorted clusters, on the other hand, show expression of *Runx1*, confirming their hematopoietic affiliation. *Cd44*, which is upregulated at the endothelial-to-hematopoietic transition (EHT) and in association with the AGM region (*Oatley et al., 2020*; *Fadlullah et al., 2022*), is detected predominantly in C-Kit+-enriched cluster 3 and CD45+-sorted clusters 1 and 4, corresponding to the later timepoints of 192 and 216 hr. This supports the notion that the later timepoints may capture AGM-like EHT, which is also suggested by significant increases in expression of *VE-cadherin* (*Cdh5*), *Sox17*, as well as *Cd34*, *Cd44*, and *Procr* (*Fadlullah et al., 2022*; *Frankish et al., 2019*; *Oatley et al., 2020*) in 192 hr, endothelial-enriched, C-Kit+ cells (*Figure 3C–D*). Important in this context, these cells express arterial endothelial markers, including *Flt1*, *Efnb2*, and *Dll4* steadily at both timepoints (*Figure 3C*, *Figure 3—figure supplement 2B*). *Hoxa9* expression also associates with arterial specification and definitive AGM-like hematopoiesis: albeit at low frequency, it is detected exclusively in C-Kit and CD45-sorted cells in the 192 and 216 hr-timepoints of haemGx differentiation (*Figure 3B and E*). These cells also express *Mllt3* and, rarely, *Hlf*, compatible with incipient specification of an AGM-like definitive blood program (*Figure 3B and E*). *Mllt3* is additionally expressed in CD41+-sorted cells of cluster 2, in line with its role in early erythroid and megakaryocytic lineage identity (*Pina et al., 2008*); *Myb* is also more frequently expressed in this population (*Figure 3E*). In contrast, *Ikzf1*, which regulates lymphoid specification (*Huang et al., 2019*), gradually increases its expression, and indeed frequency, in 216 hr-CD45+ sorted cells. A handover from erythroid/megakaryocytic to myelo-lymphoid lineage affiliation is apparent between 144 hr and 216 hr haemGx blood cells (*Figure 3—figure supplement 2C*). Of note, a small number of erythroid-affiliated cells at 216 hr express adult globins, suggesting a definitive origin (*Figure 3—figure supplement 2C*).

Altogether, scRNA-seq analysis support the suggestion from phenotypic and in vitro functional data, that the haemGx model captures two waves of hematopoietic specification from HE-like cells and may configure pre-definitive and definitive embryonic blood formation.

## Late-stage haemGx cells transcriptionally resemble mouse AGM progenitors and have short-term engraftment potential

To investigate the putative YS-like pre-definitve and AGM-like definitive nature of haemGx-generated hematopoietic progenitors, we conducted projections of haemGx scRNAseq data onto *bona fide* embryonic populations. We considered three distinct scRNA-seq datasets, which (1) capture arterial and haemogenic specification in the para-splanchnopleura (pSP) and AGM region between E8.0 and E11 (*Hou et al., 2020*; *Figure 4—figure supplement 1A*); (2) uniquely capture YS, AGM and FL progenitors and the AGM EHT (*Zhu et al., 2020*; *Figure 4—figure supplement 1B*); and (3) interrogate the mouse AGM at the point of HSC emergence from the dorsal aorta HE (*Thambyrajah et al., 2024*; *Figure 4—figure supplement 1C*). We extracted highly varying genes from each of the 3 datasets to construct dimensionality reduction maps of the data (*Figure 4—figure supplement 1A-C*), and conducted k-Nearest Neighbor (KNN) analysis to classify and regress our haemGx scRNA-seq data onto those maps considering three neighbors (*Figure 4A*). HaemGx cells with a correlation metric >0.7 were projected onto the respective embryonic datasets (*Supplementary file 3*). Projections onto *Hou et al., 2020* positioned clusters 0 and 3, corresponding to endothelium, including arterial endothelium and HE-like cells, between days E8.5 and E10, putatively spanning YS-like and AGM-like blood formation. A subset of CD45+-sorted cells, as well as a small numbers of CD41+ cells from cluster 2 projected onto E10 HE and intra-aortic cluster (IAC) cells, suggesting the presence of definitive HSPC or their precursors in haemGx (*Figure 4A*, *Supplementary file 3*). We sought to further check the transcriptional affiliation of haemGx with intra-embryonic, definitive hematopoiesis, by considering the dataset by *Zhu et al., 2020*, which uniquely captures different hematopoietic locations in the same study (*Figure 4A*). We matched the identity of a large proportion of CD41+-sorted cells to YS EMP, with some of the same cluster 2 cells mapping to erythroid progenitors, and some later CD45+-sorted cells resembling myeloid progenitors (*Figure 4A*, *Supplementary file 3*).

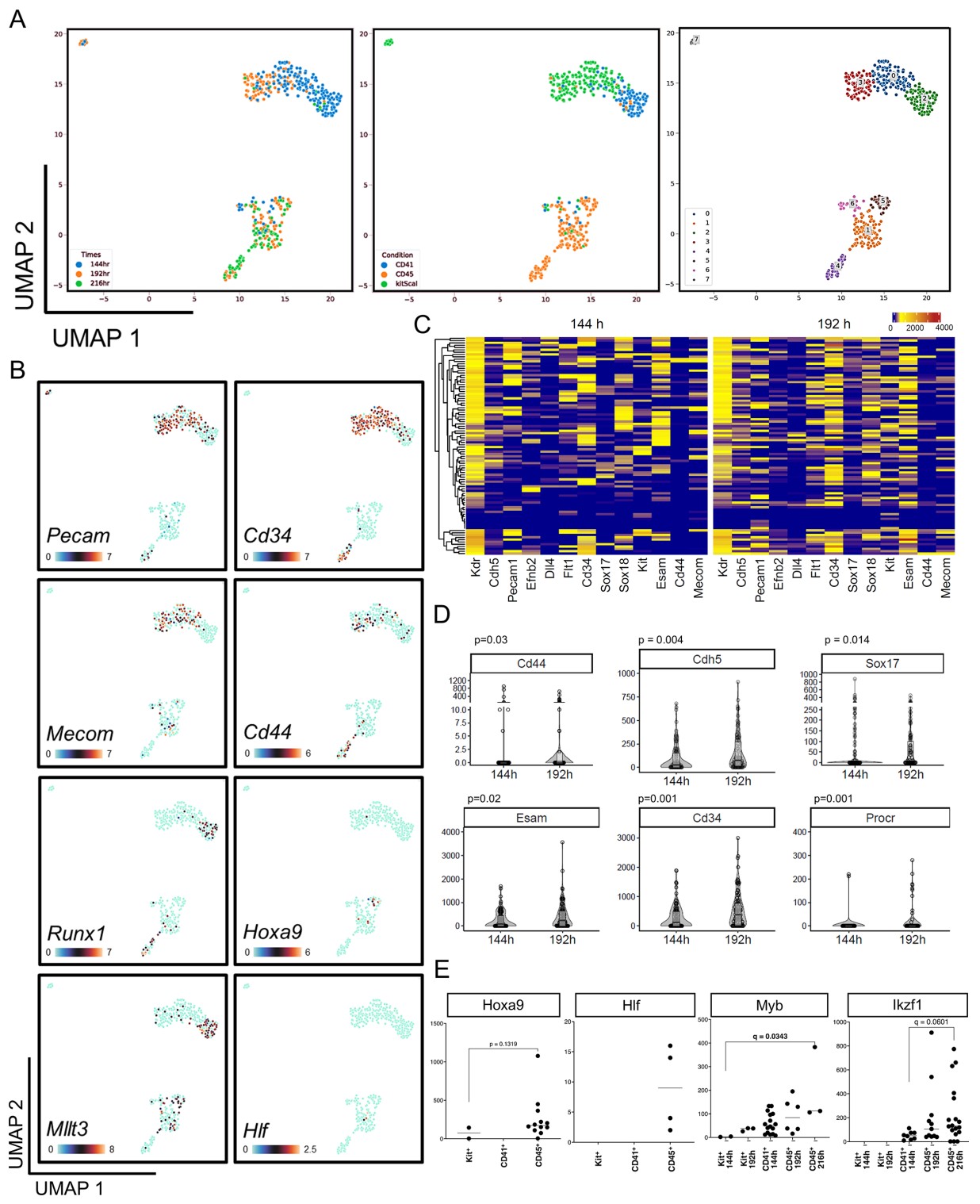

**Figure 3.** Time-resolved analysis of hemogenic gastruloid (haemGx) by scRNA-seq captures successive waves of hematopoietic specification. (**A**) UMAP clustering of subsets of sequenced cells expressing CD41, C-Kit, and CD45 at 144 hr, 192 hr, and 216 h, annotated by timepoint (left panel), sorting condition (middle panel), and Leiden clustering (right panel). (**B**) UMAP panels highlighting the expression of specific hemato-endothelial markers in the transcriptional spaces defined in (**A**). Colour scale indicates expression levels in counts. (**C**) Heatmaps of differentiation of an arterial endothelial programme in haemGx from scRNA-seq data of sorted on expression of C-Kit at 144 hr and 192 hr. Colour scale indicates expression in counts. (**D**) Violin plots quantifying the expression of hemato-endothelial markers in C-Kit+fraction from clusters in (**A**) comparing 144 hr and 192 hr timepoints. Wilcoxon test; significant p-value <0.05. (**E**) Expression of definitive hematopoietic genes in C-Kit⁺, CD41⁺ and CD45⁺ cells at 144 hr and 192/216 hr timepoints;

*Figure 3 continued on next page*

Figure 3 continued

expression levels as normalized counts per million reads. Mann-Whitney or Kruskal-Wallis with Dunn's multiple comparisons; significant p or q-value <0.05.

The online version of this article includes the following figure supplement(s) for figure 3:

**Figure supplement 1.** Single-cell RNA-seq (scRNA-seq) analysis of haemGx identifies time-dependent signatures of endothelial, hemogenic, and stromal cells.

**Figure supplement 2.** Single-cell-RNAseq (scRNAseq) analysis of haemGx.

On the other hand, a subset of CD45[+]-sorted cells mapped to IAC and lymphoid-myeloid primed progenitors (LMPP) in the FL (*Figure 4A*, *Supplementary file 3*), compatible with the presence of AGM-like definitive HSPC in haemGx. Interestingly, some CD41[+]-sorted cells also map to IAC, and indeed to HE, suggesting that this earlier population may contain HSPC precursors, in addition to YS-like progenitors. IAC and HE-like cells were also found within C-Kit[+] cells, mostly from the 192 hr timepoint, while the majority of C-Kit[+] cells captured different endothelial identities, as expected from their transcriptional profiles. Projection onto *Thambyrajah et al., 2024* (*Figure 4A*, *Supplementary file 3*), conflates YS-like and AGM-like progenitors onto the candidate HSC-enriched embryo AGM cluster 2, highlighting the similarity between transcriptional programs at different sites of embryonic blood emergence. CD45[+] cells are more distally-projected onto the same cluster, as shown for E11.5 cells in the *Thambyrajah et al., 2024* study, some of which correspond to the candidate FL-LMPP in *Zhu et al., 2020* (*Supplementary file 3*). HaemGx cell similarities to *Thambyrajah et al., 2024* HSC-enriched cluster 2 also include HE-like and IAC-like cells (as per *Zhu et al., 2020* projections), mostly E9.5-like as per *Hou et al., 2020*, overall suggesting putative early specification of the AGM region within haemGx.

## Late-stage haemGx cells can engraft hematopoietic tissues upon maturation

The presence of putative AGM-like hematopoietic cells within haemGx led us to investigate whether haemGx-derived cells had the ability to engraft the hematopoietic system of immunodeficient animals. In early experiments, we had transplanted dissociated haemGx cells obtained at 192 and 216 hr into irradiated immunocompetent C57Bl/6 recipients, sub-lethally irradiated immunodeficient NSG mice, or non-conditioned NSGW41 hosts, but failed to detect signs of engraftment of peripheral blood, spleen (Spl) or bone marrow (BM) at 12 days (including absence of spleen colony formation), at 4–6 weeks (short-term engraftment), or beyond 8–10 weeks (long-term engraftment) by either flow cytometry detection of CD45.2 or by genomic DNA (gDNA) PCR analysis of the *Flk1-GFP* locus.

Failure to engraft was perhaps not surprising given that transcriptional programs position haemGx no later than E9.5-E10, which precedes HSC specification in the AGM. Additionally, specification of HSC-supporting stroma within haemGx, namely mesenchymal cells (*Kapeni et al., 2022*) and elements of the sympathetic nervous system (*Fitch et al., 2012*), is only detectable at the 192 hr timepoint, suggesting that further 3D maturation in vivo may be required. We thus tested implantation of undissociated haemGx in the adrenal gland of *Nude* mice (*Figure 4B*), a topography more commonly used to support tumor development, but that was recently shown to be able to support HSC development, possibly with contributions from the sympathetic adrenal medullary niche (*Schyrr et al., 2023*). In order to facilitate the analysis of engrafted animals, and given the coincidence of CD45 isoform between *Flk1-GFP* mouse ES cells and the *Nude* mice in which the experiments were performed, we engineered constitutive expression of BFP from the *Rosa26* locus in the *Flk1-GFP* mES cell line (*Rosa26-BFP::Flk1-GFP*) (*Figure 4—figure supplement 2A*). We implanted 3 haemGx each unilaterally in the adrenal gland of unconditioned recipient mice (*Figure 4B*). Control mice were injected with an equivalent volume of PBS. Animals were sacrificed at 4 or 8 weeks after implantation, with 4–5 experimental animals and 1 control collected at each timepoint. We checked Spl and BM BFP engraftment, by flow cytometry and by PCR analysis of DNA. Sensitivity of PCR analysis was determined by serial dilution of *Rosa26-BFP::Flk1-GFP* mouse ES cells in the human K562 cell line, with detection of as low as 10 pg of mouse input in a total of 100 ng gDNA (*Figure 4—figure supplement 2B*), corresponding to approximately 3 cells (in 300,000). PCR analysis of BFP engraftment of the Spl and BM 4 weeks after adrenal implantation of 3 haemGx / animal detected *BFP* amplicon in

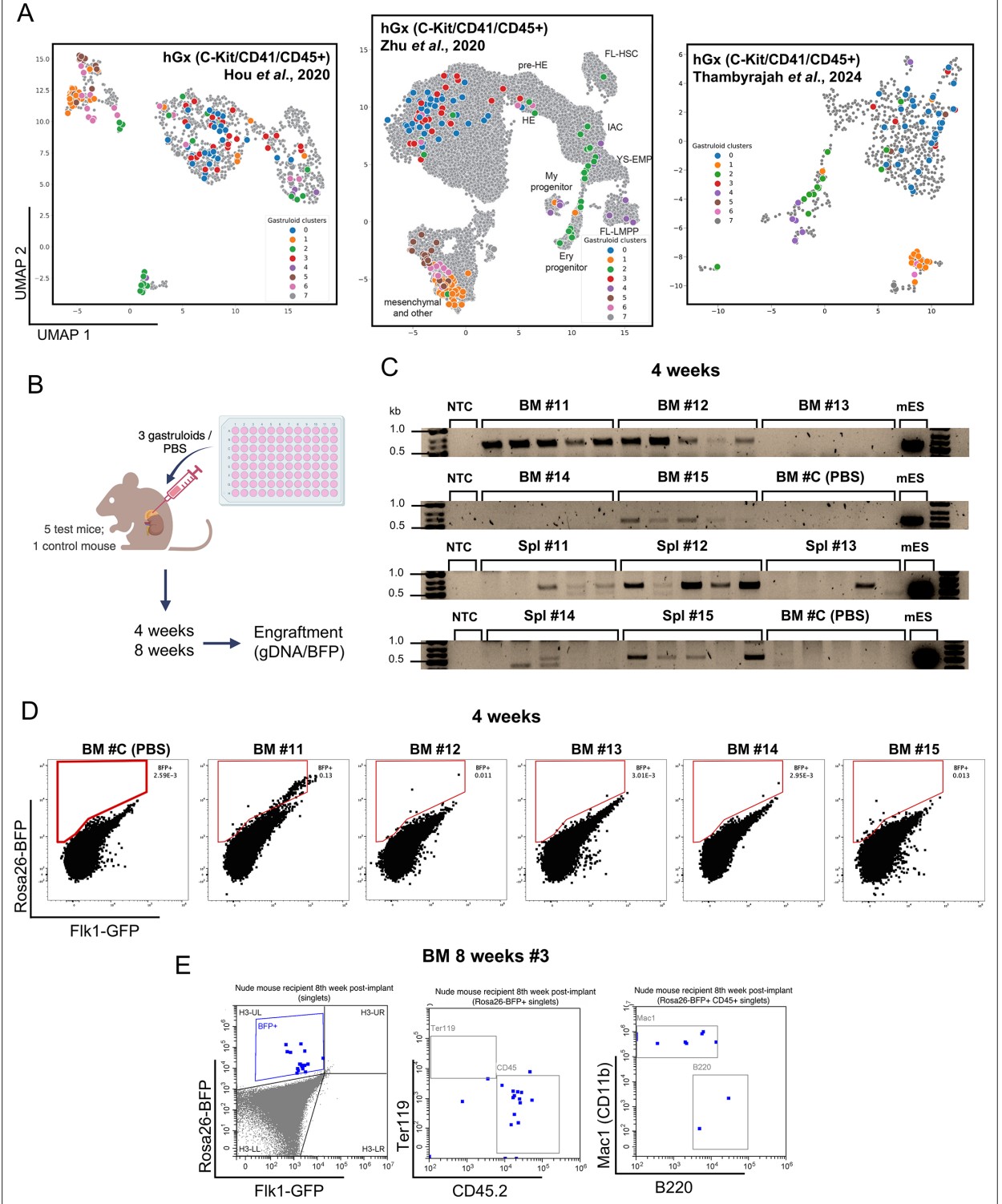

**Figure 4.** Late-stage hemogenic gastruloid (haemGx) contains hematopoietic output with transcriptional alignment to mouse embryonic populations and in vivo engraftment potential. (**A**) Projection of clustered haemGx cells (CD41, C-Kit, and CD45 at 144 hr, 192 hr, and 216 hr; *Figure 3A*) onto mouse single-cell RNA-seq datasets capturing: arterial and haemogenic specification in the para-splanchnopleura (pSP) and AGM region between E8.0 and E11 (*Hou et al., 2020*); YS, AGM, and FL progenitors and the AGM EHT (*Zhu et al., 2020*); HSC emergence from the dorsal aorta HE (*Thambyrajah et al., 2024*). Gastruloid cell projection identified as per their cluster of origin (*Figure 3A*). (**B**) Schematic representation of the experimental workflow for execution and analysis of haemGx implantation in the adrenal gland of immunodeficient mice. (**C**) PCR detection of haemGx genomic (g) DNA in the bone marrow (BM) and spleen (Spl) of immunodeficient mice 4 weeks after unilateral adrenal implantation of 3 gastruloids/gland. Analysis of

*Figure 4 continued on next page*

*Figure 4 continued*

five replicates of 100 ng gDNA/recipient tissue; control animal was injected unilaterally with PBS in an adrenal gland in parallel with experimental implantation. Reaction positive control used 100 ng of gDNA from *Rosa26-BFP::Flkj1-GFP* mES cells (mES) used to generate haemGxs. NTC: no template control. (**D**) Flow cytometry plots of engraftment detection by BFP expression in bone marrow (BM) of recipient mice 4 weeks following implantation of 216 hr haemGx. (**E**) Flow cytometry plots of lineage affiliation of BFP⁺ engraftment in bone marrow (BM) of recipient mice 8 weeks following implantation of 216 hr haemGx.

The online version of this article includes the following source data and figure supplement(s) for figure 4:

**Source data 1.** Original files for agarose gel electrophoresis in *Figure 4C*.

**Source data 2.** Annotated uncropped agarose gel electrophoresis in *Figure 4C*.

**Figure supplement 1.** Analysis of mouse embryonic scRNA-seq datasets for transcriptional similarities of haemGx outputs.

**Figure supplement 2.** Detection of engraftment potential of late-stage hemogenic gastruloid (haemGx) in adrenal glands of Nude mice.

**Figure supplement 2—source data 1.** Original files for agarose gel electrophoresis in *Figure 4—figure supplement 2B, C and E*.

**Figure supplement 2—source data 2.** Annotated uncropped agarose gel electrophoresis in *Figure 4—figure supplement 2B, C and E*.

both tissues in 3 of the 5 samples (#11, 12, and 15) (*Figure 4C*, *Figure 4—figure supplement 2C*). We sampled the gDNA of each recipient five times and detected the presence of the amplicon in 3–5 of the replicate samples of the positive animals; replicate sampling of the control was consistently negative (*Figure 4C*, *Figure 4—figure supplement 2C*). The same three animals showed BFP detection by flow cytometry above control, but no higher than 0.1% (*Figure 4D*). PCR engraftment was nevertheless detectable in both erythroid Ter119⁺ and lympho-myeloid CD45⁺ fractions of the recipient animals showed PCR engraftment in both lineage fractions (*Figure 4—figure supplement 2D and E*). Analysis of engraftment 8 weeks after implantation showed low-level detection (<0.1%) of BFP⁺ cells with positivity for CD11b⁺ myeloid and B220⁺ lymphoid compartments in BM of 1 recipient (*Figure 4D*, *Figure 4—figure supplement 2F-G*); a fourth recipient had to be sacrificed early due to infection.

Overall, implantation in the adrenal niche and/or preservation of the 3D architecture enabled low-level in vivo hematopoietic activity, although the mechanism remains unclear. In some recipients, hematopoietic engraftment was accompanied by in situ growth and multi-tissue differentiation of the implanted haemGx (*Figure 4—figure supplement 2H*). However, adrenal enlargement was not observed in all animals with hematopoietic engraftment (e.g. 4 weeks' #11 had a normally sized adrenal, as did all animals analysed at 8 weeks), making it unlikely that hematopoietic activity results from teratoma formation by residual pluripotent cells. Late-stage haemGx (216 hr) may thus contain hematopoietic progenitors with some short-term engraftment potential but incomplete functional maturation, which are present at low frequency. Engraftment is erythro-myeloid at 4 weeks and lympho-myeloid at 8 weeks, reflecting different classes of progenitors, putatively of YS-like and AGM-like affiliation.

## HaemGx resolve the early hemogenic ontogeny targeted by infant AML gene *MNX1*

The ability of haemGx to reconstitute putative YS and AGM-like HE and hemopoietic progenitors positions it as an attractive model to probe cellular origins of developmental leukemia. In particular, forms of AML unique to the first 2 years of life – infant (inf)AML – have been shown to transform fetal liver (FL) but not adult hematopoietic cells (*Waraky et al., 2024*; *Mercher et al., 2009*; *Chen et al., 2011*; *Ragusa et al., 2023*), compatible with targeting of a transient embryonic cell. In the case of t(7;12) AML (herein *MNX1-r*) (*Ragusa et al., 2023*), in which the translocation results in transformation-driving ectopic expression of *MNX1* (*Waraky et al., 2024*), analysis of patient transcriptional signatures (*Ragusa et al., 2022*) for over-representation of cell atlas-defined affiliations, indicates enrichment in cardiomyogenic, endothelial, HSC/progenitor, and mast cells (*Figure 5A*). These signature enrichments are compatible with an early hemogenic cell derivative, capturable in the haemGx model. Other cell type-enrichments reflect MNX1 functions in pancreatic and neuronal development (*Ragusa et al., 2022*; *Figure 5—figure supplement 1A*).

We overexpressed *MNX1* (MNX1-OE) in *Flk1-GFP* mouse ES cells by lentiviral transduction (*Figure 5B*) and analyzed progression of haemGx generation (*Figure 5C*). We used human *MNX1* cDNA to distinguish from the endogenous gene, but the degree of homology is nevertheless high (84%), supporting functional equivalence. We confirmed that *MNX1* overexpression in haemGx is

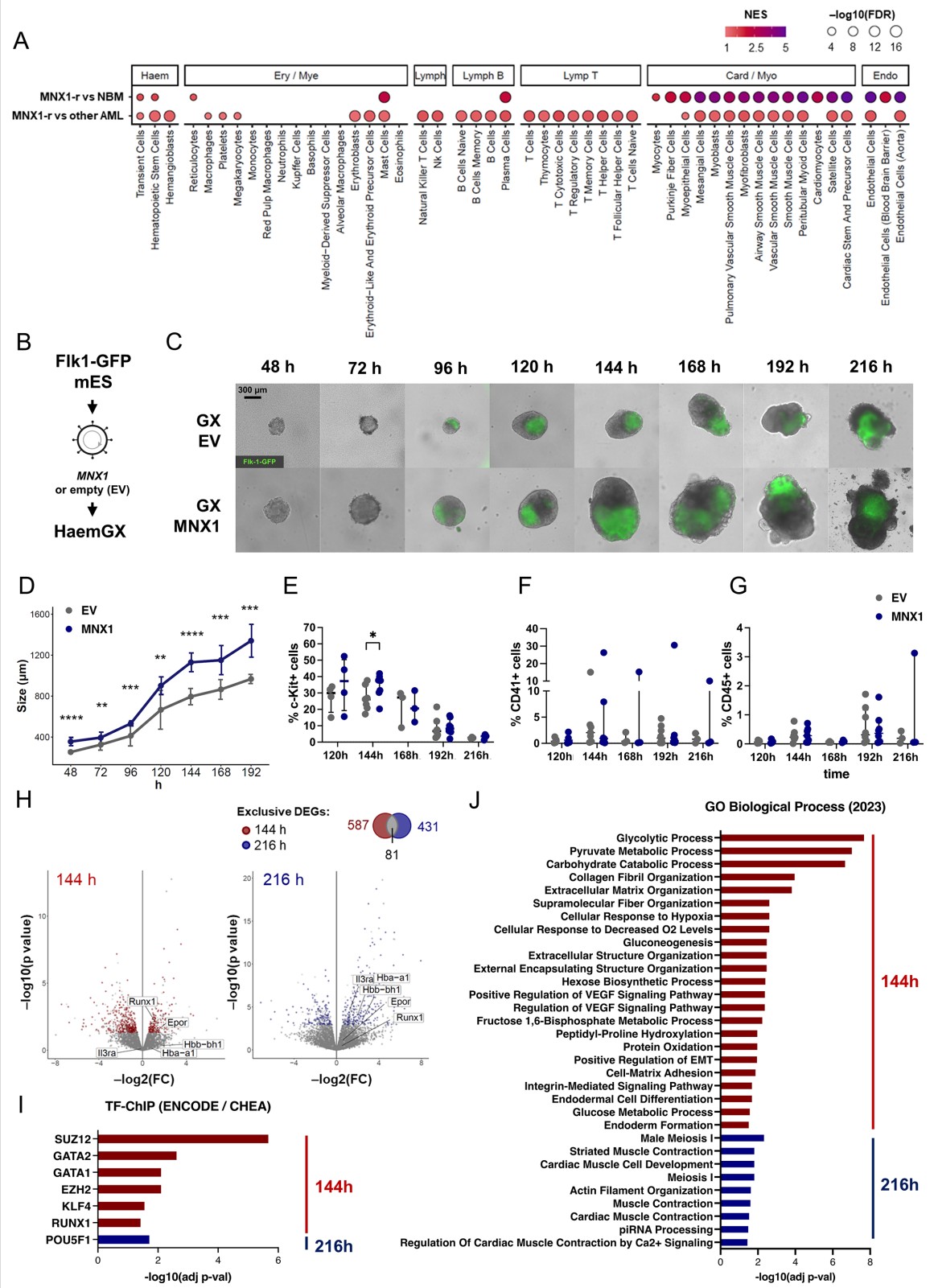

**Figure 5.** Hemogenic gastruloid (HaemGx) with *MNX1* overexpression have increased cellularity and enhanced hemogenic endothelial potential. (**A**) Cell type enrichment analysis in *MNX1-r* AML transcriptomes, compared to normal pediatric bone marrow (BM) or other pediatric AML from the TARGET database. GSEA used representative cell type gene sets from the 2021 DB database; bubble plot shows NES scores (colour gradient) and statistical significance (–log10(FDR), bubble size). (**B**) Schematic representation of generation of *MNX1*-overexpressing and empty vector (EV) mES

*Figure 5 continued*

cells by lentiviral transduction, with subsequent assembly into haemGxs. (**C**) Imaging of haemGxs with *MNX1* overexpression and EV controls at 10 x magnification, showing appropriate assembly and polarization of the Flk1-GFP marker. Scale bar: 300 mm. (**D**) Size of haemGx at each timepoint, determined by the distance between the furthest extreme points in μm. Mean ± SD of three replicate experiments; two-tailed t-test, p<0.05 (*), 0.001 (**), 0.0001 (***), and 0.00001 (****). (**E–G**) Flow cytometry timecourse analysis of (**E**) C-Kit⁺, (**F**) CD41⁺, and (**G**) CD45⁺ cell abundance in MNX1 and EV haemGxs (120–216 hr). Mean ± standard deviation of 3–7 independent experiments; two-way ANOVA and Sidak's multiple comparison test significant at p<0.05 for C-Kit⁺ cells only (construct contribution to variance p=0.0191; 144 hr comparison *p=0.0190). (**H**) Volcano plots of differentially expressed genes (DEGs) by bulk RNA-seq between MNX1 and EV haemGxs at 144 hr and 216 hr. Intersection of statistically significant DEGs between 144 hr and 216 hr (shown in Venn diagram on top right) identified unique DEGs for each condition (labelled in red for 144 hr and blue for 216 hr). (**J**) Gene Ontology (GO) term enrichment for differentially upregulated genes between MNX1 and EV haemGx at 144 h (red) and 216 h (blue), computed in EnrichR using the GO Biological Process repository. (**I**) Transcription factor (TF) binding site enrichment on differentially upregulated genes between MNX1 and EV haemGx at 144 h (red) and 216 hr (blue) using the ENCODE and ChEA Consensus TFs from ChIP database on EnrichR.

The online version of this article includes the following figure supplement(s) for figure 5:

**Figure supplement 1.** *MNX1* overexpression promotes hemogenic specification in hemogenic gastruloid (haemGxFigure 5).

**Figure supplement 2.** Transcriptional analysis of hemogenic gastruloid (haemGx) with *MNX1* overexpression.

maintained to endpoint (***Figure 5—figure supplement 1B***). MNX1-OE haemGx activated polarized *Flk1-GFP* expression and elongated with similar kinetics to empty-vector (EV) control (***Figure 5C***), albeit with formation of larger haemGx (***Figure 5D***) denoting increased cellularity (***Figure 5—figure supplement 1C***). From 192 hr onwards, MNX1-OE haemGx had a higher frequency of spontaneously contractile structures (***Figure 5—figure supplement 1D***), compatible with the cardiogenic cell association of patients' transcriptomes (***Figure 5A***).

We interrogated the time-dependent hemogenic cell composition of haemGx by flow cytometry, quantifying C-Kit, CD41, and CD45 (***Figure 5E-G***, ***Figure 5—figure supplement 1E and F***), through which we had captured endothelial/HE-like cells, YS-EMP-like, and later putative AGM-like progenitors, respectively. We observed a small but significant expansion of the C-Kit⁺ compartment specifically at 144 hr (***Figure 5E***, ***Figure 5—figure supplement 1E***). All cell markers were relatively unchanged at later timepoints, although absolute cell numbers were higher in MNX1-OE haemGx (***Figure 5—figure supplement 1C***).

To better understand the consequences of MNX1-OE on hemogenic development, we performed RNA-sequencing (RNA-seq) of haemGx at 144 hr and 216 hr, capturing putative YS and AGM-like hemopoietic waves, respectively (***Figure 5H***). We confirmed expression of the human *MNX1* transgene in the sequenced reads (***Figure 5—figure supplement 2A***); in contrast, we could not detect endogenous mouse *Mnx1* in any of the samples (***Figure 5—figure supplement 2A***), confirming that it does not normally play a role in hemogenic development. DEGs between MNX1-OE and EV (***Supplementary file 4***) configured distinct MNX1-driven programs at 144 hr and 216 hr, with >80% genes timepoint-specific (***Figure 5H***). Both programs include up-regulation of hematopoietic-associated genes, capturing EHT and progenitor regulators (e.g. *Runx1*, and *Epor* and *Zfpm1*, respectively) at the earlier timepoint, and more differentiated effectors, e.g., *Hba-a1*, *Hbb-bh1*, and *Il3ra* at 216 hr, compatible with selective targeting and/or expansion of YS EMP-like cells and their later differentiated progeny (***Figure 5H***). An overview of transcriptional regulatory programs at both timepoints denotes a clearer hemogenic enrichment at 144 hr, with over-representation of GATA2, GATA1r and RUNX1 binding targets (***Figure 5I***), putatively capturing an expansion of a HE-to-EMP-like transition, also represented in enriched GO categories related to VEGF signalling and epithelial-to-mesenchymal transition (EMT) (***Figure 5J***). In contrast, no enrichment of hemogenic transcription factor (TF)-ChIP targets or related gene ontologies (GO) categories was observed in MNX1-OE haemGx at 216 hr (***Figure 5I–J***), suggesting specific targeting of YS-like hemogenic programs. Instead, there was an enrichment in cardiogenic GO at 216 hr (***Figure 5J***), reflecting the increase in contractile foci in MNX1-OE late haemGx.

We attempted to further deconvolve the cellular heterogeneity underlying bulk RNA-seq DEGs by interrogating the cluster signatures obtained from the single-cell analysis of haemGx development (***Figure 5—figure supplement 2B and C***; refer to ***Figure 3—figure supplement 1D***). In support of an association with hemato-endothelial specification, MNX1-OE transcriptomes showed enrichment of haemGx HE cluster 5 and 0 signatures; the YS-EMP-like cluster 4 was also enriched (***Figure 5—figure supplement 2B***). MNX1-OE DEGs at both timepoints also enriched for the relatively less differentiated

cluster 10, which predominantly captures unfractionated haemGx cells at 120 hr, supporting an early ontogenic affiliation. Significantly, clusters 5 and 0 HE signatures were also enriched in *MNX1-r* infAML patient signatures (*Ragusa et al., 2022*; *Figure 5—figure supplement 2B and C*), putatively aligning MNX1-OE effects with HE and the hemogenic transition. In contrast, MNX1-OE DEGs specific to 216 hr showed enrichment in myeloid progenitor/MLP-enriched cluster 8 which captures 192/216 hr haemGx cells, but diverges from *MNX1-r* infAML signatures (*Figure 5—figure supplement 2B and C*), instead approximating the distinct myelo-monocytic and/or mixed-lineage affiliation of *KMT2A*-rearranged AML, which is also common in older pediatric and adult patients.

## Serial replating of MNX1-OE haemGx cells identifies a candidate C-Kit⁺ leukemia-propagating cell more susceptible to transformation in the YS period

We further investigated the potential of MNX1-OE haemGx to reflect *MNX1-r* AML biology by performing serial replating of CFC assays, a classical in vitro assay of leukemia transformation. We compared 144 hr and 216 hr haemGx to match functional and transcriptome data, and plated unfractionated haemGx, given the transient nature of C-Kit⁺ enrichment. Both 144 hr (*Figure 6A–B*) and 216 hr (*Figure 6C–D*) MNX1-OE haemGx replated for at least five platings were significantly different to control (EV) (*Figure 6B and D*). EV-haemGx replating could not be sustained beyond plate 3 in most experiments started from 216 h (*Figure 6C*); 144 hr EV-haemGx exhibited some low-level replating until plate 5 (*Figure 6A*), but the colonies were formed by a few scattered cells clearly distinct from the well-demarcated colonies obtained from MNX1-OE haemGx. We analyzed early-stage (plate 1) and late-stage (plate 5) colonies by flow cytometry to identify the nature of the replating cells. Cells selected through replating were C-Kit⁺ (*Figure 6E*), matching the cell phenotype transiently enriched at 144 hr. MNX1-OE C-Kit⁺ cells co-expressed CD31 (*Figure 6—figure supplement 1A and B*), underpinning the endothelial signatures of *MNX1-r* leukemia. Early replating of MNX1-OE haemGx also enriched for erythroid-affiliated Ter119⁺ cells, particularly when initiated from the 144 hr timepoint (*Figure 6—figure supplement 1C*), but this was not sustained through replating. Myelo-monocytic-affiliated markers, including CD11b or CD45, were not present (*Figure 6E*, *Figure 6—figure supplement 1C*).

We performed RNA-seq analysis of MNX1-OE haemGx CFC cells obtained through re-plating and integrated the data with the 144 hr and 216 hr haemGx timepoints to identify genes enriched through the process of transformation. Hs*MNX1* reads were increased in MNX1-OE CFC samples, supporting selective expansion of transduced cells (*Figure 7A*). We performed hierarchical clustering of all DEGs in pairwise comparisons (*Figure 7B*), and used Gene Set Enrichment Analysis (GSEA) of *MNX1-r* patient signatures up-regulated in comparison to other AML in the infant (0–2) age group, to interrogate the different clusters. Cluster k14 had significant enrichment of *MNX1-r* signatures against all other AML subtypes (*Figure 7C*), with considerable overlap between leading-edge (i.e. significantly-enriched) genes (*Figure 7—figure supplement 1A*). Importantly, k14 captures the subgroup of DEGs up-regulated upon CFC replating which are also up-regulated in MNX1-OE haemGx at 144 hr (*Figure 7B*), matching the time-dependent enrichment of the putative MNX1-OE propagating population. Interrogation of cell-type-specific signatures within the k14 *MNX1-r* leading-edge genes against the single-cell PanglaoDB atlas captured similarities to hemogenic and non-hemogenic cell types, including interneuron, and other neural signatures (*Figure 7D*), which reflect MNX1 biological functions (*Harrison et al., 1999*; *Thaler et al., 1999*; *Ragusa et al., 2022*). The most highly significant signatures capture early developmental hemogenic components, such as 'endothelial cells' and 'erythroid cell precursors,' reflecting the 144 hr, putative YS affiliation of MNX1-OE haemGx. The 'mast cells' affiliation of *MNX1-r* AML signatures within the haemGx k14 cluster is also likely to reflect a YS EMP-like affiliation (*Chia et al., 2023*), and is accordingly aligned with MNX1-OE haemGx isolated at 144 hr (*Figure 7—figure supplement 1B*). Giemsa-Wright-stained cytospins of serially replated CFC assays initiated with 144 hr MNX1-OE cells showed a shift from blast-like at plate 3 to precursor cell morphology with some differentiated mast cells at plate 5 (*Figure 7E*, *Figure 7—figure supplement 1C*). We also observed mast cells in 216 hr-initiated colonies (*Figure 7E*), but these were differentiated at an earlier plate, suggesting that MNX1-OE-responsive C-Kit⁺ progenitors experience maturation between 144 hr and 216 hr.

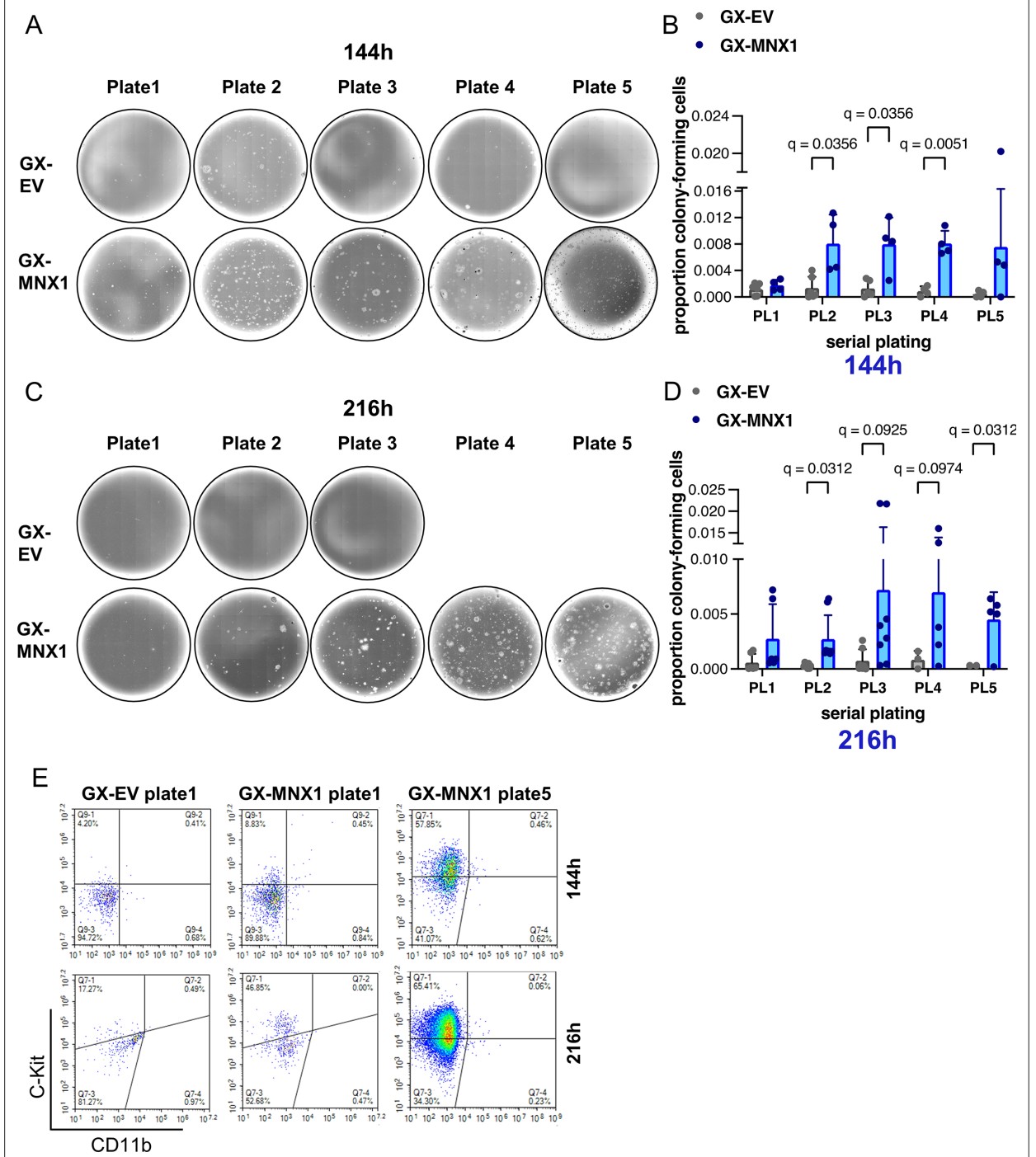

**Figure 6.** MNX1 selects C-Kit[+] clonogenic cells and selectively transforms end-stage haemGx. (**A**) Representative photographs of serial replating of colony-forming cell assays initiated from EV or MNX1 cells obtained at 144 h of the haemGx protocol. (**B**) Quantification of colony-replating efficiency of EV and MNX1 144h-haemGx (GX) cells. Mean + SD of n>3 replicates; Kruskal-Wallis with Dunn's multiple comparison testing at significant q<0.05. (**C**) Representative photographs of serial replating of colony-forming cell assays initiated from EV or MNX1 cells obtained at 216 h of the haemGx protocol. (**D**) Quantification of colony-replating efficiency of EV and MNX1 216h-haemGx cells. Mean + SD of n=5–8 replicates; Kruskal-Wallis with Dunn's multiple comparison testing at significant q<0.05. (**E**) Flow cytometry analysis of hemogenic progenitor C-Kit[+] cells and myeloid-affiliated CD11b[+] cells at first round of plating of 144 hr and 216 hr-haemGx initiated from EV and MNX1 cells; representative plots.

The online version of this article includes the following figure supplement(s) for figure 6:

**Figure supplement 1.** Characterization of *MNX1*-overexpressing replating cells from haemGx.

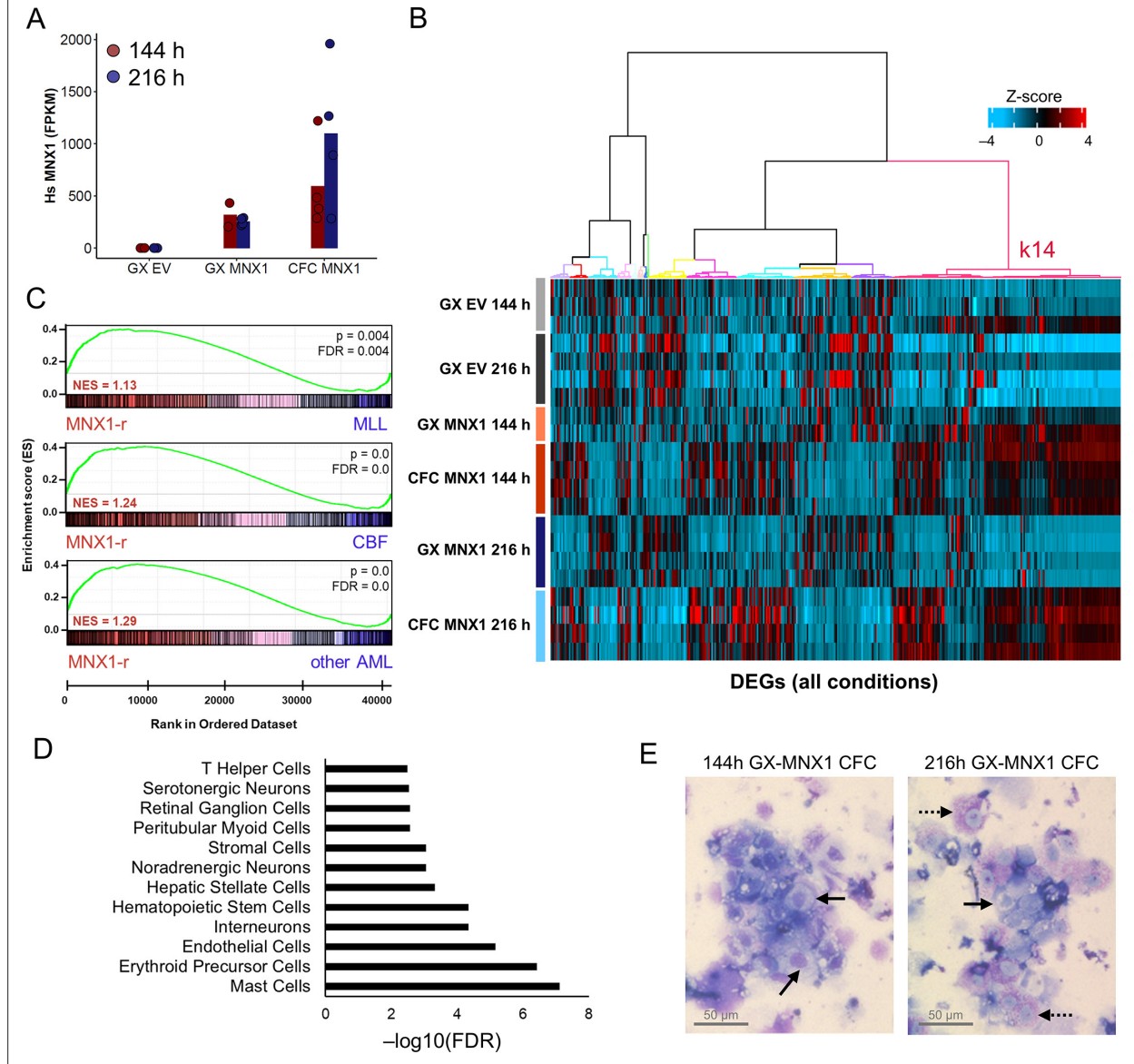

**Figure 7.** HaemGx with *MNX1* overexpression recapitulate *MNX1-r* acute myeloid leukemia patient signatures. (**A**) Quantification of human *MNX1* transcripts in FPKM units from RNA-seq of GX-EV, GX-MNX1, and CFC at 144 hr and 216 hr. Bars represent mean of replicates. (**B**) Heatmap comparing the expression of all differentially expressed genes (DEGs) between all conditions (GX-MNX1 144 hr vs GX-EV 144 hr; GX-MNX1 216 hr vs GX-EV 216 hr; CFC MNX1 144 hr vs GX MNX1 144 hr; CFC MNX1 216 hr vs GX MNX1 216 hr) as Z score. Hierarchical clustering by Ward D method on Euclidean distances identifies 14 clusters (**k**). (**C**) Gene set enrichment analysis (GSEA) plots for gene set extracted from k14 from (**B**) against *MNX1-r* patients RNA-seq counts vs MLL, core-binding factors (CBF), or other pediatric AML. (**D**) Cell type analysis of the intersect of leading-edge genes LEGs (n=476) from *Figure 7—figure supplement 1A*.in *MNX1-r* patients (k14 LEGs) (**B**) using the Panglao DB 2021 database. (**E**) Representative Giemsa-Wright stained dissociated CFC replating (fourth plating) MNX1 haemGx cells from 144 hr and 216 hr. Solid black arrows, mast cell precursors; dashed black arrows, mast cells.

The online version of this article includes the following figure supplement(s) for figure 7:

**Figure supplement 1.** Transcriptional analyses of hemogenic gastruloid (haemGx) with *MNX1* overexpression compared to patient signatures and current in vivo model.

Taken together, the data positioned the initiating cellular targets of *MNX1-r* AML at the YS endothelial-to-EMP transition. More broadly, the analysis supports the potential of haemGx to dissect the ontogeny of specific subtypes of hematological cancers initiated in utero.

## Discussion

In this study, we developed and characterized a haemGx model from mouse ES cells, which recapitulates YS-like and AGM-like waves of definitive blood progenitor specification, and has incipient ability to engraft adult hematopoietic tissues short-term upon in vivo maturation. By exploring the haemGx model through enforced expression of the *MNX1* oncogene, which is a hallmark of infant AML harbouring the t(7;12) translocation, we identified the YS endothelial-to-hematopoietic transition as the transformation target of *MNX1* gene rearrangement, shedding light on to its exclusive association with the infant age group.

Our data positions the haemGx model as an advanced platform to probe physiological and pathological development of the hematopoietic system, that benefits from the time-dependent and topographic organization associated with gastruloid models. We constructed the haemGx model by introducing a pulse of Activin A and extending the protocol under hematopoiesis-promoting cytokines, to reach the equivalent of mouse embryonic day 10 (E10) and achieve sequential acquisition of YS-like and AGM-like blood production from HE, benchmarked by surface markers CD41 and CD45, respectively. In agreement with recent observations from *Dias et al., 2025*, which highlight a critical balance between WNT-driven posterior and Nodal/TGFβ anterior signals in patterning the midbody axis and endo-mesodermal derivatives, we observed that the combination of Activin A and CHIR99021 was critical for robust *Flk1/Kdr* activation and the efficient initiation of CD41 and CD45-marked hematopoietic specification. This is in contrast with the hematopoietic output of cardiogenic gastruloids reported by *Rossi et al., 2022*, which did not require Activin A. Although our strict dependence on Activin A to achieve initial activation of *Flk1* programs and specification of a vascular network and hematopoietic programs precluded a side-by-side comparison with the Rossi protocol (*Rossi et al., 2022*), we note their apparent lower efficiency in blood formation, at least at the comparable timepoints of 144 and 168 hr, and their bias towards erythroid formation, reflecting a putative YS, and potentially primitive, nature. This bias may also be captured in the reported spleen colony formation; however, given the limited characterization of the colonies in the manuscript, which does not appraise origin, morphology, or cellularity, the nature of the spleen engraftment remains speculative. Importantly, standard gastruloid protocols with only CHIR99021 pulse and no additional cytokines (*Beccari et al., 2018*) can capture a transcriptional erythrogenic wave.

Despite low frequency engraftment of adult hematopoietic tissues by haemGx cells, we demonstrated donor origin of the engrafted cells by flow cytometry and PCR detection of a BFP transgene up to 8 weeks. Engraftment progressed from erythroid-biased at 4 weeks to myelo-lymphoid at 8 weeks, as observed in progenitor engraftment from adult cells (*Yu et al., 2024*). Critically, successful engraftment required implantation of undissociated haemGx for further maturation in a supportive microenvironment, in this case provided by the adrenal gland (*Schyrr et al., 2023*). In this study, we did not dissect the relative contribution of the continued maturation in 3D *vs* the influence of the niche for implantation, as we opted for performing the transplantation as to maximize both. Future experiments will be necessary to understand possible extrinsic niche contributions, including the sympathetic nervous system we sought to include (*Kapeni et al., 2022*; *Fitch et al., 2012*), and how they can be incorporated in the haemGx culture by extension and/or further modification of the protocol. Although we observed continued growth and maturation of the haemGx in situ after implantation, this was not the case in all animals with engraftment. On the one hand, this dissociation excludes the likelihood of the engraftment resulting from teratoma formation from the gastruloid itself. On the other hand, it raises the possibility that the inefficiency of the engraftment is the consequence of a low frequency of cells within each haemGx with HSPC properties, which we can seek to resolve by upscaling of the haemGx culture or further expansion of dissociated HSPC (*Ng et al., 2025*).

Although the system is shy of achieving HSC maturation in situ, it can mature to produce low-level engrafting multilineage progenitors, and potentially HSCs. This suggests that access to the relevant hematopoietic molecular programs is preserved. Incipient specification of non-hematopoietic tissues shown to support HSC emergence, including PDGFRA⁺ mesenchymal (*Chandrakanthan et al., 2022*) and neuro-mesenchymal (*Miladinovic et al., 2024*) cells, and neural-crest derivatives (*Kapeni et al.,*

*2022*; *Fitch et al., 2012*), occurs in sync with development of AGM-like hematopoiesis, to indicate the presence of cues that with further maturation may facilitate HSPC specification at the appropriate developmental time. Interestingly, through extending the haemGx culture, and in addition to AGM-like hematopoiesis, we also captured the potential to form renal tubular structures upon in vivo maturation, suggesting induction of lateral plate and intermediate mesoderm from which the AGM region derives, and putative reconstitution of the intra-embryonic hematopoietic niche.

Establishment of a tractable in vitro model of hematopoietic development which faithfully recapitulates time-dependent emergence of hematopoietic progenitors in their physiological niche would propel translational applications by enabling high-throughput studies of HSC emergence. Most ESC-based models of blood specification do not achieve concomitant recapitulation of the surrounding microenvironment. Protocols typically remove ESC-derived hemogenic cells from an early embryonic-like niche to direct differentiation through genetic manipulation (*Sugimura et al., 2017*) and/or empirically-determined growth factor addition (*Lis et al., 2017*; *Vo and Daley, 2015*; *Fowler et al., 2024*; *Nafria et al., 2020*; *Sroczynska et al., 2009*; *Piau et al., 2023*), with minimal probing of intermediate steps. When hematopoietic waves have been specifically inspected, emergence of YS and AGM-like stages were conflated in time (*Pearson et al., 2015*; *Murry and Keller, 2008*), suggesting a bias towards intrinsic hematopoietic cell potential, at the expense of cell fate modulation by time-coherent extrinsic physiological cues, affecting the nature of cellular output. The present haemGx system, in contrast, successfully deconvolutes YS-like and AGM-like stages of blood production, including capturing differences in the HE generation, suggesting a level of organization amenable to further development into an integrated hemogenic niche. Indeed, projections to scRNA-seq datasets of in vivo murine hematopoiesis place our endothelial and hematopoietic outputs at coherent temporal windows.

The haemGx protocol could be further developed to include more supportive structures of tissue organization, as non-hematopoietic gastruloid models have benefitted from mechanical, or chemo-mechanical, support of artificial ECM embedding (*Veenvliet et al., 2020*; *van den Brink et al., 2020*; *Dijkhuis et al., 2023*), to match temporal coherence and tissue polarization, with detailed tissue architecture illustrated by somite formation. Prior to HSC emergence in the AGM, critical signals such as SCF and Shh, which we have delivered to the cultures, are secreted by different cells on opposite dorsal-ventral locations (*Souilhol et al., 2016*), indicating the potential relevance of tissue organization to improve and potentially extend hematopoietic formation. Moreover, HSPC formation in the AGM is facilitated by circulatory shear stress (*North et al., 2009*; *Diaz et al., 2015*; *Lundin et al., 2020*), suggesting that introduction of flow may improve hemato-endothelial organization and boost blood production.

We attested the utility of haemGx's to interrogate the biology of leukemias with developmental origins, thanks to a nearly full recapitulation of embryonic hemogenic specification, including critical early stages of HE production and EHT to EMP and MLP/MPPs. We modelled *MNX1-r* AML, a poorly characterized form of leukemia exclusive to the infant age group (*Ragusa et al., 2023*). Overexpression of *MNX1* in the haemGx model propagated cells bearing transcriptional resemblance to patient *MNX1-r* AML, overall positioning the haemGx as an attractive tractable model in which to study developmental leukemias. We showed that MNX1-OE transiently expanded a C-Kit$^+$ cell with hemato-endothelial characteristics at the YS-like stage, and could selectively sustain the phenotype through serial replating of CFC assays. Replating CFCs could be initiated beyond the YS-like stage, albeit at the relative expense of an early differentiation state and of the similarity to patient transcriptome, putatively defining a window of susceptibility to MNX1-driven transformation.

Importantly, the haemGx model was able to resolve the cellularity and transcriptional features inferred from patient transcriptomes, with the advantage of a higher resolution in dissecting the contribution of cell-of-origin in the appropriate developmental context.

A recent paper using MNX1-OE has shown that MNX1 can initiate a transplantable leukemia in immunodeficient mice by targeting FL, but not BM, cells (*Waraky et al., 2024*). This suggests that ectopic expression of *MNX1* transforms an embryonic cell, rather than depends on an embryonic niche. It also suggests that MNX1-OE can cause leukemia as a single genetic hit, although it cannot be excluded that other mutations are required for, and may indeed have contributed to, full transformation in vivo, as observed in a recent report confirming the in utero origin of the leukemia (*Bousquets-Muñoz et al., 2024*). However, the mouse model does not address the nature of the embryonic cell

targeted by MNX1-OE. Like the MNX1-OE haemGx cells selected through serial replating, MNX1-OE mouse leukemia are also C-Kit⁺ (*Waraky et al., 2024*). Interestingly, clusters of genes up-regulated in the mouse MNX1-OE leukemia (*Figure 7—figure supplement 1D* – k1, 2, 5, 6, and 9) do not robustly separate *MNX1-r* from other infAML by GSEA analysis (*Figure 7—figure supplement 1E*), and have diverged more from the patient data in their cell-type signature analysis than the original MNX1-OE cells in the FL (*Figure 7—figure supplement 1F*), putatively selecting a more generic program of leukemia transformation. With the resolution provided by the haemGx model, we propose that the divergence from *MNX1-r* infAML signatures increases from MNX1-OE haemGx 144 hr, to 216 hr to FL, positioning the origin of the AML at the YS stage, likely at the HE-to-EMP transition. This cell may persist in the haemGx culture system to the 216 hr timepoint, and migrate to the FL in the embryo, eventually extinguishing through differentiation, as suggested by the MNX1-OE haemGx CFC assays. Persistence of the original MNX1-OE target cell will determine the sensitivity to MNX1-OE, and can be addressed in the haemGx system through lineage tracing. Whether additional mutations, namely a reported association between *MNX1-r* and tri(19) (*Espersen et al., 2018*; *Ragusa et al., 2023*), contribute to prolonging the undifferentiated state of the candidate YS target cell with increased probability of transformation, or otherwise expand the targeted cell in a more classical second-hit fashion, can potentially also be screened in the haemGx system, in combination with transplantation experiments.

Over the last decade, the effort to achieve robust hematopoietic production from PSC, mouse and human, has been centred on the generation of HSCs, a critical translational need (*Chang et al., 2023*; *Dijkhuis et al., 2023*; *Fidanza and Forrester, 2021*). There are good indications that this is achievable, both with (*Vo and Daley, 2015*; *Dang et al., 2002*) and without (*Piau et al., 2023*) genetic manipulation. However, success remains episodic, mainly because mechanistic understanding of processes underpinning blood generation during development is limiting, exacerbated by limiting access to human embryonic and fetal material. Ongoing optimization of gastruloid models and the success at integration of organoid cultures in microfluidic systems (*Saorin et al., 2023*; *Quintard et al., 2024*) will allow us to further optimize haemGx culture. In conclusion, the haemGx represents a faithful in vitro reference to replace and complement embryo studies, holding promise to understand the biology of healthy and malignant developmental hematopoiesis and advance blood production for clinical use.

## Materials and methods

**Key resources table**

| Reagent type (species) or resource | Designation | Source or reference | Identifiers | Additional information |
|---|---|---|---|---|
| Strain, strain background (*E. coli*) | NEB 5-alpha Competent *E. coli* | New England Biolabs (NEB) | Cat. #C2987H | Competent cells |
| Antibody | Anti-mouse c-Kit (CD117) clone 2B8 – APC-Cy7 — Monoclonal, Rat | BioLegend | Cat. #105825 | Flow cytometry (1:100) |
| Antibody | Anti-mouse CD41 clone mwreg30 – PE-Dazzle594 — Monoclonal, Rat | BioLegend | Cat. #133935 | Flow cytometry (1:100) |
| Antibody | Anti-mouse CD45 clone 30F11 – APC — Monoclonal, Rat | BioLegend | Cat. #103111 | Flow cytometry (1:100) |
| Antibody | Anti-mouse CD144 (VE-cadherin) clone BV13 – APC — Monoclonal, Rat | BioLegend | Cat. #138011 | Flow cytometry (1:100) |
| Antibody | Anti-mouse CD43 clone S11 – PE — Monoclonal, Rat | BioLegend | Cat. #143205 | Flow cytometry (1:100) |
| Antibody | Anti-mouse Ter119 – PC7 — Monoclonal, Rat | BioLegend | Cat. #116223 | Flow cytometry (1:100) |

*Continued on next page*

*Continued*

| Reagent type (species) or resource | Designation | Source or reference | Identifiers | Additional information |
|---|---|---|---|---|
| Antibody | Anti-mouse CD31 – biotin — Monoclonal, Rat | BD Pharmingen | Cat. #553371 | Flow cytometry (1:100) |
| Antibody | Anti-mouse CD34 – APC — Monoclonal, Armenian Hamster | BioLegend | Cat. #128611 | Flow cytometry (1:100) |
| Antibody | CD11b anti-mouse/human clone M1/70 – biotin — Monoclonal, Rat | BioLegend | Cat. #101203 | Flow cytometry (1:100) |
| Antibody | Mouse Anti-Mouse CD45.2 clone 104 – PE — Monoclonal, Mouse | BD Biosciences | Cat. #560695 | IF (1:200) |
| Antibody | Rat Anti-Mouse CD31 clone MEC13.3 – biotin — Monoclonal, Rat | BD Biosciences | Cat. #553371 | IF (1:200) |
| Antibody | Goat Anti-Mouse c-Kit/ CD117 — Polyclonal, Goat | R&D Systems | Cat. #AF1356SP | IF (1:200) |
| Peptide, recombinant protein | Murine LIF | Peprotech | Cat. #250–02 | Medium supplement |
| Peptide, recombinant protein | Stemmacs pd0325901 | Miltenyi biotec | Cat. #130-106-5411 | Medium supplement |
| Peptide, recombinant protein | Chiron (chir99021) | Biogems | Cat. #2520691 | Medium supplement |
| Peptide, recombinant protein | Activin A Plus | Qkine | Cat. #qk005 | Medium supplement |
| Peptide, recombinant protein | Murine Vegf 165 | Peprotech | Cat. #450–32 | Medium supplement |
| Peptide, recombinant protein | Murine Fgf-basic | Peprotech | Cat. #450–33 | Medium supplement |
| Peptide, recombinant protein | Murine Sonic Hedgehog (Shh) | Peprotech | Cat. #315–22 | Medium supplement |
| Peptide, recombinant protein | Murine Tpo | Peprotech | Cat. #315–14 | Medium supplement |
| Peptide, recombinant protein | Murine Flt3-ligand | Peprotech | Cat. #250–31 l | Medium supplement |
| Peptide, recombinant protein | Murine Scf | Peprotech | Cat. #250–03 | Medium supplement |
| Other | N2B27 medium (NDiff 227) | Takara bio | Cat. #y40002 | Medium |
| Other | DMEM/F-12 | Gibco | Cat. #11320033 | Medium |
| Other | Neurobasal medium | Gibco | Cat. #21103049 | Medium |
| Other | N-2 supplement | Gibco | Cat. #17502048 | Medium |
| Other | B-27 supplement | Gibco | Cat. #17504044 | Medium |
| Other | Mouse methylcellulose complete media | R&D systems | Cat. #hsc007 | Medium |
| Cell line (*M. musculus*) | *Kdr(Flk1)*-GFP mouse embryonic stem cell line | Alfonso Martinez Arias Lab; *Jakobsson et al., 2010* | | Maintained in ESLIF medium |

*Continued on next page*

*Continued*

| Reagent type (species) or resource | Designation | Source or reference | Identifiers | Additional information |
|---|---|---|---|---|
| Cell line (*M. musculus*) | *Sox17*-GFP mouse embryonic stem cell line | Alfonso Martinez Arias Lab; *Niakan et al., 2010* | | Maintained in ESLIF medium |
| Cell line (*M. musculus*) | *T/Bra*::GFP mouse embryonic stem cell line | Alfonso Martinez Arias Lab; *Fehling et al., 2003* | | Maintained in ESLIF medium |
| Cell line (*M. musculus*) | E14Tg2a mouse embryonic stem cell line | MMRRC, University of California Davis, US; *Hooper et al., 1987* | | Maintained in ESLIF medium |
| Cell line (*M. musculus*) | *Kdr*(*Flk1*)-GFP::Rosa26-BFP mouse embryonic stem cell line | This paper | N/A | Materials and Methods (Cell culture), *Figure 4—figure supplement 2A* |
| Cell line (*H. sapiens*) | HEK293T | ATCC | CRL-3216 | Maintained in DMEM-F12, 10% FBS, 1% P/S |
| Sequence-based reagent | MNX1 forward | *Gulino et al., 2021* | qPCR primers | 5-GTTCAAGCTCAACAAGTACC-3 |
| Sequence-based reagent | MNX1 reverse | *Gulino et al., 2021* | qPCR primers | GGTTCTGGAACCAAATCTTC-3 |
| Sequence-based reagent | Ppia forward | *Moris et al., 2018* | qPCR primers | TTACCCATCAAACCATTCCTTCTG-3 |
| Sequence-based reagent | Ppia reverse | *Moris et al., 2018* | qPCR primers | AACCCAAAGAACTTCAGTGAGAGC-3 |
| Sequence-based reagent | BFP forward | This paper | PCR primers | 5-GCACCGTGGACAACCATCACTT-3 |
| Sequence-based reagent | BFP reverse | This paper | PCR primers | 5-CAGTTTGCTAGGGAGGTCGC-3 |
| Recombinant DNA reagent | Pwpt-lssmorange-PQR | Ghevaert Lab, WT-MRC Cambridge Stem Cell Institute, UK *Dalby et al., 2018* | Vector | |
| Recombinant DNA reagent | Pwpt-LssmOrange-PQR-MNX1 | Cloned by Biomatik Corporation (Kitchener, Canada) | Vector | |
| Recombinant DNA reagent | Pcdna3-Rosa26-Rosa26-BFP | This paper | Vector | Construct described in *Figure 4—figure supplement 2A*; Materials and methods (Lentiviral vector packaging and transduction) |
| Other | CELLSTAR cell-repellent cell culture plate, 96 well, U-bottom | Greiner (Bio-One) | Cat. #650970 | |

## Cell culture

*Kdr*(*Flk1*)-GFP (*Jakobsson et al., 2010*), *Kdr*(*Flk1*)-GFP::Rosa26-BFP, Sox17-GFP (*Niakan et al., 2010*), T/Bra::GFP (*Fehling et al., 2003*), and E14Tg2A (*Hooper et al., 1987*) mouse embryonic stem cells (mES) lines were cultured in ES + LIF medium in gelatinized flasks (0.1% gelatin) with daily medium change, as previously described (*Turner et al., 2017*). The ES + LIF medium contained 500 ml Glasgow MEM BHK-21 (Gibco), 50 ml of fetal bovine serum (FBS, Embryonic Stem Cells tested, Biosera, Nuaillé, France), 5 ml GlutaMAX supplement (Gibco), 5 ml MEM Non-Essential Amino Acids (Gibco), 5 ml Sodium pyruvate solution (100 mM, Gibco), and 1 ml 2-Mercaptoethanol (50 mM, Gibco). Murine LIF (Peprotech) was added at 1000 U/ml. HEK293T cells were grown in Dulbecco's modified Eagle medium (Gibco) supplemented with 10% FBS (Gibco) and 1% penicillin/streptomycin (Gibco). All cultures were kept at 37 °C and 5% $CO_2$. Cell lines tested negative for Mycoplasma contamination by PCR. The presence of the relevant constructs in cell lines was tested throughout the manuscript.

## Hemogenic gastruloids assembly, culture, and dissociation

mES were maintained in ES + LIF medium and transferred to 2i+LIF (containing Chiron and MEK inhibitor PD03) for 24 hr prior to the assembly into gastruloids. 250 untransduced mES cells (or 400 for GX-MNX1 and GX-EV) were seeded in each well of a U-bottom, cell-repellent 96-well plate (Greiner Bio-One, Stonehouse, UK) in 40 µl of N2B27 medium [Takara Bio or home-made as Cold Spring Harbour Protocols 'N2B27 Medium' (2017)]. The plate was centrifuged at 750 rpm for 2 min to promote deposition and aggregation of the cells and was then incubated at 37 °C, 5% CO2 for 48 hr. After 48 hr, 150 µl of N2B27 medium supplemented with 100 ng/ml Activin A (QKine, Cambridge, UK) and 3 µM chiron (Peprotech) was added to each well. At 72 hr, 150 µl of medium were removed, without disrupting the gastruloids in the wells. 100 µl of N2B27 with 5 ng/ml of Vegf and Fgf2 each (Peprotech) were added to each well. From 72 hr to 144 hr, each day 100 µl of medium were removed and replaced with N2B27+Vegf + Fgf2. At 144 h, the medium was further supplemented with Shh at 20 ng/ml. From 168 hr to 216 hr, the medium was N2B27+5 ng/ml Vegf, plus 20 ng/ml mTpo, 100 ng/ml mFlt3l, and 100 ng/ml mScf (Peprotech). To dissociate cells from the gastruloid structures, medium was removed and individual gastruloids were collected using a pipette and precipitated at the bottom of a microcentrifuge tube. The remaining medium was aspirated and the bulk of gastruloids was washed in PBS. 50 µl of TrypLE express was added to pelleted gastruloids to be incubated at 37 °C for 2 min to dissociate cells.

## Animals and implantation

Female athymic nude-Foxn1[nu] (nu/nu) mice (Envigo, Bresso, Italy) were housed under pathogen-free conditions. In accordance with the '3Rs policy,' experiments were reviewed and approved by the Animal Welfare Body (OPBA) of IRCCS Ospedale Policlinico San Martino and by the Italian Ministry of Health (n. 883/2020-PR).

Intact haemGx (3 per mouse in 10 µL of PBS) were inoculated in the adrenal gland of five-week-old mice. Briefly, mice were anesthetized with a mixture of xylazine (ROMPUN, Bayer) and ketamine (LOBOTOR, Acme S.r.l.) and injected with haemGx, after laparotomy, in the capsule of the left adrenal gland, as previously described (*Brignole et al., 2023*; *Pastorino et al., 2003*). All mice survived surgery. Control mice (CTR) received vehicle only. Mice body weight and general physical conditions were recorded daily. One month after haemGx inoculation, mice were sacrificed and spleen and bone marrow (BM) from femurs collected. BM cells and spleens, mechanically reduced to cell suspension, were stored at –80 °C until use. Adrenal glands were paraffin-embedded for subsequent IHC studies.

## Methylcellulose colony-forming assays (CFC)

Disassembled gastruloids cells were plated on Mouse Methylcellulose Complete Media (R&D Systems). Cells were first suspended in Iscove's Modified Dulbecco's Medium (IMDM, Gibco) with 20% FBS (Gibco) before addition to the methylcellulose medium. Cells were plated in duplicate 35 mm dishes with $2\times10^5$ cells/plate. Plates were incubated at 37 °C and 5% CO2 for 10 days, when colonies were scored. For serial replating experiments, cells in methylcellulose were collected and washed in phosphate buffer saline (PBS) to achieve single-cell suspensions and replated as described above. Dissociated gastruloid cells from CFC were collected as cell suspensions in PBS and centrifuged onto slides (Shandon Cytospin 2 Cytocentrifuge) at 350 rpm for 5–7 min, with a PBS-only pre-spin at 1000 rpm, 2 min. Slides were stained in Giemsa-Wright stain for 30 s followed by addition of phosphate buffer, pH 6.5 for 5 min.

## Immunofluorescence staining

Immunostaining of whole gastruloids was performed as described before (*Baillie-Johnson et al., 2015*). Briefly, gastruloids were fixed in 4% paraformaldehyde (PFA) dissolved in PBS for 4 hr at 4 °C on orbital shaking and permeabilized in PBSFT (10% FBS and 0.2% Triton X-100), followed by 1 hr blocking in PBSFT at 4 °C on orbital shaking. Antibody dilutions were made in PBSFT at 1:200 for primary and 1:500 for secondary antibodies. Antibody incubations were performed overnight at 4 °C on orbital shaking, and subjected to optical clearing overnight in ScaleS4 clearing solution. Individual gastruloids were then mounted on glass coverslips by pipetting as droplets in ScaleS4 and DAPI nuclear stain.

## Histology and H&E staining

Samples were fixed with 10% formalin at room temperature for 48 hr followed by an ethanol series of ethanol and paraffin; ethanol 70% (90 min), ethanol 80% (120 min), ethanol 95% (180 min), three times ethanol 100% of 120, 120, and 180 min, 50% ethanol/paraffin (3×120 min) and three times paraffin (60, 90, and 120 min) before embedding in histological cassettes. Blocks were sectioned with a HistoCore BIOCUT Microtome (Leica) at 10 mm and slides were stained with Haematoxylin/Eosin.

## Imaging

Images of cultured gastruloids and CFC plates were captured using the Cytation 5 Cell Imaging Multi-Mode Reader (Biotek) plate reader using bright field and FITC channels. ImageJ (*Schneider et al., 2012*) was used for gastruloid size quantification. Confocal microscopy was performed on LSM700 on a Zeiss Axiovert 200 M with Zeiss EC Plan-Neofluar 10 x/0.30 M27 and Zeiss LD Plan-Neofluar 20 x/0.4 M27 objective lens. Tissue sections were imaged with a THUNDER Imager 3D microscope system (Leica).

## Lentiviral vector packaging and transduction

The lentiviral overexpression vector pWPT-LSSmOrange-PQR was used to clone the *MNX1* gene cDNA. The viral packaging vectors pCMV and pMD2.G, described in *Pina et al., 2008*, were used to assemble lentiviral particles using HEK293T cells via transfection using TurboFect Reagent (Invitrogen). Transduction of mES cells was performed overnight by addition of lentivirus to culture medium and washed the following day (*Moris et al., 2018*). Transduced cells were sorted for positivity to LSSmOrange.

## Flow cytometry

Surface cell marker analysis was performed by staining using the antibodies listed in key resources table. Disassembled gastruloids cells were resuspended in PBS, 2% FBS, and 0.5 mM EDTA and stained at a dilution of 1:100 for primary antibodies for 20 min at 4 °C. When indicated, streptavidin was added at a dilution of 1:200. Analysis was performed on ACEA Novocyte (Agilent) or Attune NxT (Thermo) analyzers, using the respective software packages. Cell sorting was performed using a CS&T calibrated BDFACS Aria III system (488 nm 40 mW, 633 nm 20 mW, 405 nm 30 mW, and 561 nm 50 mW), set with the 100 μm nozzle at 20 PSI and a 4-way purity mask or, in the case of haemGx adrenal implants, a Beckman Coulter CytoFlex SRT (488 nm 50 mW, 633 nm 100 mW, 405 nm 90 mW, and 561 nm 30 mW), set with 100 μm nozzle at 15 PSI, also using 4-way purity. Single-cell deposition in 96-well plates was performed using single-cell sorting mode. Intact cells were gated on FSC-A vs SSC-A plot, followed by doublet exclusion on FSC-A vs FSC-H and SSC-A vs SSC-H, prior to gating on fluorescent parameters for the markers described in the results.

## Single-cell RNA sequencing of time-resolved hemogenic gastruloids

Gastruloids were collected at different timepoints of the protocol, disassembled, and FACS deposited into 96-well plates, either as unsorted (global) or sorted by CD45, CD41, and C-Kit/Sca1 markers (sorted). RNA from the cells were extracted from single cells using Smart-seq2 technology at 500 Kb and depth of 151 bases. Sequencing reads were quality-checked using FastQC (v0.11.9). We trimmed the samples using trimGalore! (0.6.6) with a cutoff of 30, clipping 15 base pairs and retaining reads of more than 100 bases. Alignment was performed on STAR (2.7.8 a) and *Mus musculus* annotations from GENCODE vM26. Aligned BAM files were annotated using featureCounts (v2.0.1) and count matrices were computed in Python by directly accessing the BAM files with gtfparse packages and collapsing lanes.

For quality control, based on the histogram of counts and multimodality distributions, we set a minimum count threshold of 200,000 counts, a minimum threshold of 1000 expressed genes, and a maximum threshold of 20% mitochondrial fraction per cell. We performed the same procedure of the second dataset with a more restrictive threshold of 400,000 counts and similar expressed genes and mitochondrial thresholds. Cells that did not pass the quality control metrics were omitted from analysis. We normalized the cells to the mean count number per dataset and applied a plus-one-log transformation of the data before proceeding to the downstream analysis.

Dimensionality reduction was performed on feature selection of the gene space using the function scanpy.highly_varying_genes with default parameters. Selection of principal components in principal component analysis (PCA) was performed by heuristic elbow method. Nearest neighbor analysis was constructed by KNN graphs using a correlation metric and 10 nearest neighbors. Data was projected for low-dimensional visualization using the UMAP algorithm with default parameters as implemented in scanpy.tl.umap. We used the Leiden algorithm as implemented in scanpy.tl.leiden to partition the data into clusters. In order to assess the election of the resolution parameter, we used Newman-Girvan modularity as a metric of clustering quality. Differentially expressed genes were computed comparing each cluster against the rest using the Wilcoxon test with Benjamini-Hochberg correction.

### Annotation and projection to additional datasets

Raw count matrices were downloaded and processed following the same pipeline as described above for scRNA-seq of gastruloids to generate UMAP and clustering. We implemented the scmap algorithm to compare our scRNA sequencing of gastruloids with the available datasets. We reduced the dimensionality of the space by selecting highly varying genes from the annotated dataset. Then, we constructed a KNN classifier with correlation metric and computed the nearest neighbors of the target data. If neighbors with correlation metrics below 0.7 default standards, the projected cells were not projected onto any cell from the annotated dataset. To visualize the cells over the UMAP plots of the other datasets, we constructed a KNN regressor with three neighbors and a correlation metric. Confusion matrices were used to visualize the overlap between gastruloid clusters and the embryo dataset assigned clusters using scmap.

### Bulk RNA sequencing

Total RNA was extracted from disassembled gastruloid cells at 216 hr. Sequencing was performed on NovaSeq PE150 platform, at 20 M paired-end reads per sample. Tophat2 with Bowtie2 were used to map paired-end reads to the reference *Mus musculus* genome build GRCm39. GENCODE Release M30 (*Frankish et al., 2019*) was used as the reference mouse genome annotation. Aligned reads were filtered by quality using samtools (*Li et al., 2009*) with a minimum threshold set at 30 (q30). Transcript assembly and quantification was achieved using htseq (*Putri et al., 2022*). Differential expression between sample and control was performed by collapsing technical replicates for each condition using DESeq2 (*Love et al., 2014*) in R environment (Deseq2 library v 1.32.0). The differential expression was expressed in the form of log2 fold change and filtered by false discovery rate (FDR) of 0.05.

### Gene list enrichment analyses

Gene ontology (GO) analysis was performed in ExpressAnalyst (available at https://www.expressanalyst.ca) using the PANTHER or GO Biological Process (BP) repository. GO terms and pathways were filtered by false discovery rate (FDR) with a cut-off of ≤0.1 for meaningful association. EnrichR (*Xie et al., 2021*; *Kuleshov et al., 2016*; *Chen et al., 2013*) was used for cell type analysis using the Panglao DB (*Franzén et al., 2019*) databases using a FDR threshold of ≤0.1 or p-value ≤0.01, where specified, as well as transcription factor (TF) binding site enrichment using the ENCODE and ChEA Consensus TFs from ChIP database.

### Gene set enrichment analysis (GSEA)

Custom gene signatures were used as gene sets for GSEA analysis (*Subramanian et al., 2005*) on the GSEA software v4.2.3 on RNA sequencing expression values in counts units. GSEA was ran in 10,000 permutations on gene set using the weighted Signal2Noise metric. Enrichment metrics are shown as normalized enrichment score (NES) and filtered by FDR ≤0.05. Leading-edge genes (LEGs) are genes with a 'Yes' values for core enrichment. For AML patient analysis, clinical phenotype and expression data (in counts units) were extracted from the GDC TARGET-AML cohorts in the Therapeutically Applicable Research to Generate Effective Treatments project (TARGET, https://ocg.cancer.gov/programs/target), downloaded from the University of California Santa Cruz (UCSC) Xena public repository (last accessed 31st August 2022). Patient samples were selected according to the reported karyotype to include t(7;12), inv(16), MLL, normal karyotype, and t(8;21). GSEA was performed comparing RNA sequencing counts of t(7;12) samples against pooled AML subtypes (inv(16), MLL, normal karyotype, and t(8;21)) as 'other AML'.

### Real-time polymerase chain reaction (qPCR)

Extracted RNA was reverse-transcribed into complementary DNA (cDNA) using High-Capacity RNA-to-cDNA Kit (Applied Biosystems). QPCR was performed using FastGene 2 x IC Green Universal qPCR Mix (Nippon Genetics, Duren, Germany) using primers for human *MNX1* (forward 5-GTTCAAGC TCAACAAGTACC-3; reverse 5-GGTTCTGGAACCAAATCTTC-3) (*Gulino et al., 2021*) and *Ppia* for endogenous control (forward 5-TTACCCATCAAACCATTCCTTCTG-3; reverse 5-AACCCAAAGAAC TTCAGTGAGAGC-3) (*Moris et al., 2018*), or Taqman gene expression assays (Mm00501741_m1 *Myb*; Mm00439364_m1 *Hoxa9*; Mm00723157_m1 *Hlf*; Mm01213404_m1 *Runx1*; Mm00444619_m1 *Dll4*; Mm00491303_m1 *Mecom*; Mm02342430_g1 *Ppia*) using Taqman Gene Expression Mastermix (Applied Biosystems). Differential gene expression was calculated using the delta delta Ct ($\Delta\Delta$Ct) method.

### Polymerase chain reaction (PCR)

Genomic DNA was extracted using Monarch Genomic DNA Purification Kit (New England Biolabs, Hitchin, UK) using manufacturer's instructions. PCR on genomic DNA was performed using Phire PCR Master Mix (Thermo Scientific) to amplify the BFP locus using forward primer 5'-GCACCGTGGACA ACCATCACTT-3' and reverse primer 5'-CAGTTTGCTAGGGAGGTCGC-3'.

### Statistical analysis

Experiments were performed at least in triplicates, unless specified otherwise. Data are plotted to include standard deviation (+/-SD) between replicates. Statistical significance was set at a threshold of p-value <0.05. Statistical analysis was performed in R environment (version 4.1.3) or using GraphPad Prism 8.0 software and is detailed in respective figure legends.

## Acknowledgements

This project was funded by a start-up grant and a BRIEF award from Brunel University London, a Little Princess Trust (LPT) Project Grant (CCLGA 2023 22 Pina) and a National Centre for the Replacement, Refinement and Reduction of Animals in Research (NC3Rs) project grant (NC/Z500677/1) to CP, by an Agencia Estatal de Investigación grant (PLEC2021-007518) to AB and AMA, and by an ERC Advanced Grant (MiniEmbryoBlueprint 834580) to AMA. LPT research is funded in partnership with CCLG through the CCLG Charity Research Network. DR received funding from the Royal Society of Biology (MRSB Travel Grant). GTC was funded by grant FPU18/05091 from the Spanish Ministry of Universities. AJ is funded by a Lady Tata Memorial Trust International Scholarship (2022). SvdB was funded by an EMBO Postdoctoral Fellowship (ALTF 195–2021) and is in receipt of a HFSP Postdoctoral Fellowship (LT0047/2022 L). Work in the CP lab was also funded by a KKLF Intermediate Fellowship (KL888), a Leuka John Goldman Fellowship for Future Science (2017–2019), and a Wellcome Trust/ISSF Bridge Funding award at the University of Cambridge (2019). JGO acknowledges financial support from the Spanish Ministry of Science and Innovation and FEDER (grant PGC2018-101251-B-I00), by the Maria de Maeztu Programme for Units of Excellence in R&D (grant CEX2018-000792-M), and by the Generalitat de Catalunya (ICREA Academia programme). MP acknowledges funding from the Italian Ministry of Health (Ricerca Finalizzata 5 per mille and Ricerca Corrente to MP). Library preparation and next-generation sequencing for single-cell RNA-seq analysis were performed by the Single Cell Genomics Group at the National Centre for Genomic Analysis – Centre for Genomic Regulation (CNAG-CRG), Barcelona. The authors wish to acknowledge Tina Balayo, Ana Filipa Domingues, Shikha Gupta, Oliver Davies, Kristen Place, and Remisha Gurung's technical support at different stages of the project.

## Additional information

### Competing interests

Alfonso Martínez Arias: AMA is an inventor in two patents on Human Polarised Three-dimensional Cellular Aggregates PCT/GB2019/052670 and Polarised Three-dimensional Cellular Aggregates PCT/GB2019/052668. The other authors declare that no competing interests exist.

## Funding

| Funder | Grant reference number | Author |
|---|---|---|
| Brunel University London | BRIEF Award Pina | Cristina Pina |
| Little Princess Trust | CCLGA 2023 22 Pina | Cristina Pina |
| National Centre for the Replacement Refinement and Reduction of Animals in Research | NC/Z500677/1 | Cristina Pina |
| Agencia Estatal de Investigación | PLEC2021-007518 | Anna Bigas Alfonso Martínez Arias |
| European Research Council | MiniEmbryoBlueprint 834580 | Alfonso Martínez Arias |
| Royal Society of Biology | MRSB Travel Grant Ragusa | Denise Ragusa |
| Ministerio de Universidades | FPU18/05091 | Gabriel Torregrosa Cortes |
| Lady Tata Memorial Trust | Memorial Trust International Scholarship 2022 | Ayona Johns |
| European Molecular Biology Organization | ALTF 195-2021 | Susanne van den Brink |
| Human Frontier Science Program | LT0047/2022-L | Susanne van den Brink |
| Kay Kendall Leukaemia Fund | KL888 | Cristina Pina |
| Leuka | Leuka John Goldman Fellowship for Future Science 2017-2019 | Cristina Pina |
| Wellcome Trust | ISSF Bridge Funding Award 2019 | Cristina Pina |
| Ministerio de Ciencia e Innovación | PGC2018-101251-BI00 | Jordi Garcia-Ojalvo |
| Agencia Estatal de Investigación | CEX2018-000792-M | Jordi Garcia-Ojalvo |
| Generalitat de Catalunya | ICREA Academia | Jordi Garcia-Ojalvo |
| Ministero della Salute | Ricerca Corrente | Mirco Ponzoni |
| Ministero della Salute | Ricerca Finalizzata 5 per mille | Mirco Ponzoni |

The funders had no role in study design, data collection and interpretation, or the decision to submit the work for publication. For the purpose of Open Access, the authors have applied a CC BY public copyright license to any Author Accepted Manuscript version arising from this submission.

## Author contributions

Denise Ragusa, Data curation, Formal analysis, Validation, Investigation, Visualization, Methodology, Writing – original draft, Writing – review and editing; Chun Wai Suen, Formal analysis, Validation, Investigation, Methodology; Gabriel Torregrosa Cortes, Data curation, Software, Formal analysis, Investigation, Visualization, Methodology, Writing – review and editing; Fabio Pastorino, Resources, Investigation, Methodology, Writing – review and editing; Ayona Johns, Validation, Investigation; Ylenia Cicirò, Investigation, Visualization, Writing – review and editing; Liza Dijkhuis, Connor Byrne, Giulia-Andreea Ionescu, Investigation; Susanne van den Brink, Investigation, Writing – review and editing; Michele Cilli, Joana Cerveira, Kamil R Kranc, Methodology; Victor Hernandez-Hernandez, Investigation, Methodology; Mirco Ponzoni, Resources, Methodology; Anna Bigas, Resources, Funding acquisition, Writing – review and editing; Jordi Garcia-Ojalvo, Resources, Supervision, Funding acquisition, Writing – review and editing; Alfonso Martínez Arias, Conceptualization, Resources, Funding

acquisition, Writing – review and editing; Cristina Pina, Conceptualization, Resources, Formal analysis, Supervision, Funding acquisition, Investigation, Visualization, Writing – original draft, Project adminis-tration, Writing – review and editing

### Author ORCIDs

Denise Ragusa (ID) https://orcid.org/0000-0002-0303-8683
Gabriel Torregrosa Cortes (ID) https://orcid.org/0000-0003-4528-5663
Ylenia Cicirò (ID) http://orcid.org/0000-0003-1607-3266
Anna Bigas (ID) https://orcid.org/0000-0003-4801-6899
Alfonso Martínez Arias (ID) https://orcid.org/0000-0002-1781-564X
Cristina Pina (ID) https://orcid.org/0000-0002-2575-6301

### Ethics

In accordance with the "3Rs policy", experiments were reviewed and approved by the Animal Welfare Body (OPBA) of IRCCS Ospedale Policlinico San Martino and by the Italian Ministry of Health (n. 883/2020-PR).

Reviewer #1 (Public review): https://doi.org/10.7554/eLife.102324.3.sa1
Reviewer #2 (Public review): https://doi.org/10.7554/eLife.102324.3.sa2
Reviewer #3 (Public review): https://doi.org/10.7554/eLife.102324.3.sa3
Author response https://doi.org/10.7554/eLife.102324.3.sa4

---

## Additional files

### Supplementary files

Supplementary file 1. Cluster classifier gene lists for each cluster by differential gene expression analysis of scRNA-seq of hemogenic gastruloid (haemGx) (all conditions), obtained by Wilcoxon rank test of each cluster against all other clusters.

Supplementary file 2. Cell type enrichment analysis of cluster classifier genes, showing the inferred identity of clusters of scRNA-seq of hemogenic gastruloid (haemGx) (all conditions). Classifier genes (in *Supplementary file 1*) were obtained by differential expression in comparison to all other clusters and subjected to enrichment analysis using the PanglaoDB repository (*Franzén et al., 2019*).

Supplementary file 3. Summary of projections of scRNA-seq of hemogenic gastruloid (haemGx) cells onto available datasets by *Hou et al., 2020*, *Zhu et al., 2020*, and *Thambyrajah et al., 2024*, showing the identity of each haemGx cell projecting to annotated clusters of the respective studies.

Supplementary file 4. List of differentially expressed genes between hemogenic gastruloid (haemGx)-MNX1 and haemGx-MNX1 at 144 hr and 216 hr compared to empty vector (EV) control.

MDAR checklist

### Data availability

Raw data as well as processed count matrices and post-processed files from single cell RNA-seq for the time-resolved data is available at at Array Express with accession code E-MTAB-12148. Bulk RNA-seq of MNX1 overexpressing gastruloids is available at Array Express with accession code E-MTAB-12173. The post-processing was performed in Python on DockerHub: dsblab/single_cell_analysis:0.5. Scripts are available in https://github.com/dsb-lab/blood_gastruloids (copy archived at *Torregrosa Cortes, 2025*) and Zenodo (https://doi.org/10.5281/zenodo.7053423). The results published here are partly based upon data generated by the Therapeutically Applicable Research to Generate Effective Treatments (TARGET) (https://ocg.cancer.gov/programs/target) initiative, of the Acute Myeloid Leukemia (AML) cohort GDC TARGET-AML. The data used for this analysis are available at https://portal.gdc.cancer.gov/ and https://xenabrowser.net/.

The following datasets were generated:

| Author(s) | Year | Dataset title | Dataset URL | Database and Identifier |
|---|---|---|---|---|
| Ragusa D, Suen W, Cortés GT, Pina C | 2022 | Single cell analysis of gastruloid hemogenic development | https://www.ebi.ac.uk/biostudies/arrayexpress/studies/E-MTAB-12148 | ArrayExpress, E-MTAB-12148 |
| Ragusa D, Suen W, Cortés GT, Pina C | 2022 | RNA sequencing of mouse embryonic stem cells (mES) and hemogenic gastruloids with MNX1 overexpression | https://www.ebi.ac.uk/biostudies/ArrayExpress/studies/E-MTAB-12173?query=E-MTAB-12173 | ArrayExpress, E-MTAB-12173 |

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
