## [Editor Report · eLife Assessment]

This **valuable** study presents a mouse gastruloid system to generate successive waves of hematopoietic progenitors that in vivo would emerge during embryonic development. Although this newly revised manuscript has addressed some of the concerns raised during the first round of review, the study is still considered **incomplete**, as the claims are only partially supported. In particular, the claim of definitive wave hematopoietic progenitors being produced in the gastruloids, and their engraftment after transplantation, would benefit from further validation.

---

## [Referee Report · Reviewer #1 (Public review)]

Summary

This manuscript describes a haemogenic gastruloid system that the authors claim recapitulates early mouse embryonic development to produce sequential waves of yolk sac and AGM-like haematopoiesis, with spatial and temporal accuracy. The model claims to reproduce mouse development to 'beyond' the E9.0 stage and apply its use to the aetiology of infant leukaemia.

Strengths

Gastruloids models are useful systems for studying early embryonic development, recapitulating aspects of gastrulation, anteroposterior regionalisation and somitogenesis. Gastruloid models that specifically mimic particular regions of the embryo could provide insights into how these regions form during development.

Weaknesses

There are a couple of major issues with this manuscript that I feel need to be addressed.

Firstly, the authors acknowledge that the proportion of blood cells that are produced by their haemogenic gastruloid system is very low - there are fewer than 2% of either blood or endothelium produced. The authors argue however, that this is because they have developed a hematopoietic organoid that captures much more of the essence of the developing embryo and therefore has a broader tissue representation and a more relevant spatial representation.

In order to prosecute this argument, this reviewer needs to understand how the differentiation protocol achieves this end, ie what is notable about the combination of factors and other media components. Also, they need to know what the evidence is to support this claim, in other words, what are the tissues that make up the organoid and is it truly representative of what would be expected in a developing embryo over this time. Does it pass from epiblast to primitive streak and then to cells of the germ layers? And how do haemGXs at different times map onto the developing mouse embryo?

Secondly, the point is repeatedly made by the authors that the distinction between non-engrafting yolk sac hematopoiesis and AGM-like hematopoiesis from which repopulating HSCs first derive is not really possible without spatial cues. This is really not true. It has been shown by a number of investigators, and summarised in a recent review (Abuhantash et al 2021), that the expression of HOXA cluster genes - most prominently HOXA9 - clearly distinguishes AGM-derived, from yolk sac derived cells. In this manner, it is evident from the UMAP provided that the is no HOXA9 expressed in either endothelium or blood cells. This argues very strongly against the proposition that AGM-type hematopoiesis is generated. Indeed, given the duration of the organoid culture of only 9 days (216hrs), it would be highly unlikely that development would even reach the stage of AGM hematopoiesis (E11.5 in the mouse), even with a 1:1 concordance between embryonic time and in vitro differentiation. Finally, if there is recapitulation of the normal pattern of embryogenesis, it would be expected that there would be a prominent phase of yolk sac hematopoiesis antedating AGM-associated hematopoiesis, which should be observed in the haemGx.

I feel that these are major conceptual points that need to be addressed in this manuscript.

---

## [Referee Report · Reviewer #2 (Public review)]

Summary:

In this manuscript, the authors describe the development of a new hemogenic gastruloid (hGX) system, which they claim recapitulates the sequential generation of various hematopoietic cell types. A key proposed advantage of this system is its ability to more faithfully model the spatiotemporal emergence of hematopoietic progenitors within a physiologically relevant niche, as compared to existing in vitro platforms. While the authors provide some initial characterisation and demonstrate the utility of the system in studying infant leukemia, the presented data are not fully conclusive and fall short of robustly supporting several of their key assertions.

Strengths:

The development of a novel in vitro system to model hematopoietic development is innovative and could potentially address important limitations of existing platforms.

Weaknesses:

The characterization of the hematopoietic progenitors generated by the hGX system is not fully convincing. The evidence supporting the emergence of late yolk sac (YS) progenitors, including lymphoid cells, and AGM-like pre-hematopoietic stem cells (pre-HSCs), is incomplete and relies heavily on transcriptomic profiling and a limited set of markers.

The identification of lymphoid or pre-HSC-like populations is primarily inferred from scRNA-seq data. The lack of robust functional validation (e.g., lymphoid differentiation assays or long-term repopulation experiments) significantly weakens the manuscript's main claims.

In the revised manuscript, the authors incorporate single-cell RNA-seq analyses indicating that their cells resemble AGM-derived endothelial-to-hematopoietic transition (EHT) populations. However, they do not test whether these cells might more closely resemble YS-derived EHT, which remains an unresolved and critical question. Additionally, the claim in line 263 that Cluster 8 (CD45⁺ cells at 192-216 h) expresses lymphoid markers is not clearly supported by the provided supplemental data (Supplemental File S1-S2).

While the authors respond that they did not claim to generate bona fide HSCs, they do state at the end of the Introduction (lines 116-121) that their system captures AGM hematopoiesis. The current data do not support this conclusion and instead suggest that the system recapitulates the generation of multipotent lymphoid progenitors (MLPs) akin to those found in the YS.

The engraftment data presented are not particularly convincing. It is unclear why the analysis was terminated at 8 weeks post-transplant, especially given that a minimum of 12 weeks is generally required to meaningfully assess the presence of pre-HSCs or bona fide HSCs with long-term repopulating potential.

Given the uncertainties discussed above, the interpretability of the MNX1 overexpression study is limited.

The authors could have more directly tested their claim of capturing multiple hematopoietic waves by performing kinetic analyses of colony-forming potential, with the expectation that more multipotent colonies would emerge at later time points. Additionally, isolating and characterizing the potential of hemogenic endothelium at different time points corresponding to the putative waves would have strengthened their conclusions. In the absence of such data, it remains unclear whether the system recapitulates sequential waves of hematopoiesis or merely reflects the progressive maturation of cells originating from a single wave.

---

## [Referee Report · Reviewer #3 (Public review)]

The authors present a revised version of their manuscript (Ragusa et al.) describing a hemogenic gastruloid (haemGx) model, used to investigate stages of blood production in vitro and for modeling a rare type of infant leukemia. The revisions address several major concerns raised during the initial round of review, and new data have been provided that overall improve the clarity and rigour of the study. In particular, the additional flow cytometry, single-cell RNA-seq analyses, and benchmarking against in vivo datasets help, to some extent, to substantiate the claims of developmental relevance of haemGx to yolk sac (YS)- and AGM-like hematopoietic waves. Nonetheless, some issues remain, particularly regarding the claims of short-term engraftment, novelty of the model, and the extent to which AGM-like HSPC are truly captured.

Major Points:

(1) The authors have clarified the novelty of their haemGx protocol relative to existing gastruloid models, including the importance of the Activin A pulse and protocol extension to 216h. Flow cytometry and scRNA-seq analyses support the emergence of endothelial and hematopoietic populations with dynamic marker expression. However, direct side-by-side comparisons with previously published protocols (e.g., Rossi et al., 2022) remain limited. The claim of "spatio-temporal accuracy" should be more cautiously phrased.

(2) The characterization of the identity of the hematopoietic waves generated in the haemGx system has been improved in the revised manuscript. Flow cytometry analysis now includes CD31/CD34 co-expression in CD41+ and CD45+ subsets, and scRNA-seq re-clustering supports two hematopoietic waves with distinct marker sets (e.g., Gata2/Myb vs. Hoxa9/Ikzf1). Projection onto multiple embryonic reference datasets (Hou et al., Zhu et al., Thambyrajah et al.) is a valuable addition. The case for YS-like EMP and AGM-like HSPC precursors is reasonably made, though further functional distinctions (e.g., lineage output differences) would strengthen the claims.

(3) The authors have now provided additional evidence for low-level engraftment following adrenal implantation of whole haemGx. Although technically demanding, this in vivo result remains marginal and should be interpreted with caution. Crucially, this still does not demonstrate HSC-level repopulation capacity. The revised manuscript has softened the claims accordingly, now referring to "progenitor" activity rather than "pre-HSC." We agree that this adjusted claim is more suitable, though the reproducibility of this experiment is still unclear.

(4) The MNX1 overexpression experiments are generally convincing in showing early expansion of a putative HE-to-EMP-like population and transcriptional resemblance to MNX1-r AML. However, the evidence for transformation is still solely based on in vitro data and lacks any evidence of in vivo leukaemia engraftment. The ability to perturb the system would add translational value to the haemGx platform, although future studies are needed to better define transformation dynamics and leukemogenic progression.

---

## [Author Response]

The following is the authors’ response to the original reviews.

**Reviewer #1 (Public review):**
SummaryThe authors describe a method for gastruloid formation using mouse embryonic stem cells (mESCs) to study YS and AGM-like hematopoietic differentiation. They characterise the gastruloids during nine days of differentiation using a number of techniques including flow cytometry and single-cell RNA sequencing. They compare their findings to a published data set derived from E10-11.5 mouse AGM. At d9, gastruloids were transplanted under the adrenal gland capsule of immunocompromised mice to look for the development of cells capable of engrafting the mouse bone marrow. The authors then applied the gastruloid protocol to study overexpression of Mnx1 which causes infant AML in humans.In the introduction, the authors define their interpretation of the different waves of hematopoiesis that occur during development. 'The subsequent wave, known as definitive, produces: first, oligopotent erythro-myeloid progenitors (EMPs) in the YS (E8-E8.5); and later myelo-lymphoid progenitors (MLPs - E9.5-E10), multipotent progenitors (MPPs - E10-E11.5), and hematopoietic stem cells (HSCs - E10.5-E11.5), in the aorta-gonad-mesonephros (AGM) region of the embryo proper.' Herein they designate the yolk sac-derived wave of EMP hematopoiesis as definitive, according to convention, although paradoxically it does not develop from intra-embryonic mesoderm or give rise to HSCs.

Our definition of primitive and definitive waves is widely used in the field (e.g. PMID: 18204427; PMID: 28299650; PMID: 33681211). Definitive haematopoiesis, encompassing EMP, MLP, MPP and HSC, highlights their origin from haemogenic endothelium, generation of mature cells with adult characteristics from progenitors with multilineage potential and direct and indirect developmental contributions to the intra-embryonic and time-restricted generation of HSCs.

General commentsThe authors make the following claims in the paper:(1) The development of a protocol for hemogenic gastruloids (hGx) that recapitulates YS and AGMlike waves of blood from HE.(2) The protocol recapitulates both YS and EMP-MPP embryonic blood development 'with spatial and temporal accuracy'.(3) The protocol generates HSC precursors capable of short-term engraftment in an adrenal niche.(4) Overexpression of MNX1 in hGx transforms YS EMP to 'recapitulate patient transcriptional signatures'.(5) hGx is a model to study normal and leukaemic embryonic hematopoiesis.There are major concerns with the manuscript. The statements and claims made by the authors are not supported by the data presented, data is overinterpreted, and the conclusions cannot be justified. Furthermore, the data is presented in a way that makes it difficult for the reader to follow the narrative, causing confusion. The authors have not discussed how their hGx compares to the previously published mouse embryoid body protocols used to model early development and hematopoiesis. Specific points(1) It is claimed that HGxs capture cellularity and topography of developmental blood formation. The hGx protocol described in the manuscript is a modification of a previously published gastruloid protocol (Rossi et al 2022). The rationale for the protocol modifications is not fully explained or justified. There is a lack of novelty in the presented protocol as the only modifications appear to be the inclusion of Activin A and an extension of the differentiation period from 7 to 9 days of culture. No direct comparison has been made between the two versions of gastruloid differentiation to justify the changes.

The Reviewer paradoxically claims that the protocol is not novel and that it differs from a previous publication in at least 2 ways – the patterning pulse and the length of the protocol. Of these, the patterning pulse is key. As documented in Fig. 1S1, we cannot obtain Flk1-GFP expression in the absence of Activin A (Fig. 1S1A), and the concentration of Activin A scales activity of the Flk1 locus (Fig. 1S1B). Expression of Flk1 is a fundamental step in haemato-endothelial specification and, accordingly, we do not see CD41 or CD45+ cells in the absence of Activin A. Furthermore, these markers also titrate with the dose of Activin A (in Fig. 1S1B).

Also, in our hands, there is a clear time-dependent progression of marker expression, with sequential acquisition of CD41 and CD45, with the latter not detectable until 192h (Fig. 1C-D), another key difference relative to the Rossi et al (2022) protocol. We suggest, and present further evidence for in this rebuttal and the revised manuscript, that the 192h-timepoint captures the onset of AGM-like haematopoiesis. We have edited the manuscript to clarify the differences and novelty in our protocol (lines 132-143) and provided a more detailed comparison with the report from Rossi et al. (2022) in the Discussion (lines 574-586).

The inclusion of Activin A at high concentration at the beginning of differentiation would be expected to pattern endoderm rather than mesoderm. BMP signaling is required to induce Flk1+ mesoderm, even in the presence of Wnt.

Again, we call the Reviewer’s attention to Fig. 1S1A which clearly shows that Activin A (with no BMP added) is required for induction of Flk1 expression, in the presence of Wnt. Activin A in combination with Wnt, is used in other protocols of haemato-endothelial differentiation from pluripotent cells, with no BMP added in the same step of patterning and differentiation (PMID: 39227582; PMID: 39223325). In the latter protocol, we also call the Reviewer’s attention to the fact that a higher concentration of Activin A precludes the need for BMP4 addition. Finally, one of us has recently reported that Activin A, on its own, will induce Flk1, as well as other anterior mesodermal progenitors (https://www.biorxiv.org/content/10.1101/2025.01.11.632562v1). In addressing the Reviewer’s concerns with the dose of Activin A used, we titrated its concentration against activation of Flk1, confirming optimal Flk1-GFP expression at the 100ng/ml dose used in the manuscript. We have included this data in the manuscript in Figure 1S1B.

FACS analysis of the hGx during differentiation is needed to demonstrate the co-expression of Flk1GFP and lineage markers such as CD34 to indicate patterning of endothelium from Flk1+ mesoderm. The FACS plots in Fig. 1 show C-Kit expression but very little VE-cadherin which suggests that CD34 is not induced. Early endoderm expresses C-Kit, CXCR4, and Epcam, but not CD34 which could account for the lack of vascular structures within the hGx as shown in Fig. 1E.

We were surprised by the Reviewer’s comment that there are no endothelial structures in our haemogenic gastruloids. The presence of a Flk1-GFP+ network is visible in the GFP images in Fig. 1B, from 144h onwards, and is detailed in the revised Fig. 2A, which shows overlap between Flk1GFP and the endothelial marker CD31. In addition, our single-cell RNA-seq data, included in the manuscript, confirms the presence of endothelial cells with a developing endothelial, including arterial, programme. This is now presented in the revised Fig. 3B-D of the manuscript, which updates a representation in the original manuscript. In contrast with the Reviewer’s claims that no endothelial cells are formed, the data show that Kdr (Flk1)+ cells co-express Cdh5/VE-Cadherin and indeed Cd34, attesting to the presence of an endothelial programme. Arterial markers Efnb2, Flt1, and Dll4 are present. A full-blown programme, which also includes haemogenic markers including Sox17, Esam, Cd44 and Mecom is clear at early (144h) and, particularly at late (192h) timepoints in cells sorted on detection of surface C-Kit (Fig. 3B-E in the manuscript). To address the specific point by the Reviewer, we also document co-expression of Flk1-GFP, CD34 and/or CD31 by flow cytometry (Fig. 2S1A-B in the revised manuscript).

To summarise new and revised data in the manuscript in relation to this point:

Immunofluorescence staining showing the Flk1-GFP-defined vascular network in Figure 1E and co-expression of endothelial marker CD31 in Figure 2A. In text: lines 159-163; 178-180.

Flow cytometry analysis of co-expression of Flk1-GFP with CD31 and CD34 in Figure 2S1AD, including controls. In text: 180-187.

Real-time quantitative (q)PCR analysis showing time-dependent expression of haematoendothelial and arterial markers in Figure 2F (specifically Dll4 and Mecom). In text: 200-209.

An improved representation of our scRNA-seq data highlighting key haemato-endothelial markers in Figure 3B-D. In text: 268-304

(2) The protocol has been incompletely characterised, and the authors have not shown how they can distinguish between either wave of Yolk Sac (YS) hematopoiesis (primitive erythroid/macrophage and erythro-myeloid EMP) or between YS and intraembryonic Aorta-Gonad-Mesonephros (AGM) hematopoiesis. No evidence of germ layer specification has been presented to confirm gastruloid formation, organisation, and functional ability to mimic early development. Furthermore, differentiation of YS primitive and YS EMP stages of development in vitro should result in the efficient generation of CD34+ endothelial and hematopoietic cells. There is no flow cytometry analysis showing the kinetics of CD34 cell generation during differentiation. Benchmarking the hGx against developing mouse YS and embryo data sets would be an important verification.

The Reviewer is correct that we have not provided detailed characterisation of the different germ layers, as this was not the focus of the study. In that context, we were surprised by the earlier comment assuming co-expression of C-Kit, Cxcr4 and Epcam, which we did not show, while overlooking the endothelial programme reiterated above, which we have presented. Given our focus on haemato-endothelial specification, we have started the single-cell RNA-seq characterisation of the haemogenic gastruloid at 120h and have not looked specifically at earlier timepoints of embryo patterning. This said, we show the presence of neuroectodermal cells in cluster 9; on the other hand, cluster 7 includes hepatoblast-like cells, denoting endodermal specification (Supplementary File S2). However, in the absence of earlier timepoints and given the bias towards mesodermal specification, we expect that specification of ectodermal and endodermal programmes may be incomplete.

In respect of the contention regarding the capture of YS-like and AGM-like haematopoiesis, we had presented evidence in the original version of the manuscript that haemogenic cells generated during gastruloid differentiation, particularly at late 192h and 216h timepoints project onto highly purified CKit+ CD31+ Gfi1-expressing cells from mouse AGM (PMID: 38383534), providing support for at least partial recapitulation of the corresponding developmental stage. These projections are represented in Fig. 4A, right and 4S1C of the revised manuscript. In distinguishing between YS-like and AGM-like haematopoiesis, we call the Reviewer’s attention to the replotting of the single-cell RNA-seq data already in the manuscript, which we provided in response to point 1 (Fig. 3B-D and 3S2B), which highlights an increase in Sox17, but not Sox18, expression in the 192h haemogenic endothelium, which suggests an association with AGM haematopoiesis (PMID: 20228271). A significant association of Cd44 and Procr expression with the same time-point (Fig. 3B-D in the manuscript), further supports an AGM-like endothelial-to-haematopoietic transition at the 192h timepoint. We have re-analysed the scRNA-seq data to better represent the expression of these markers in Fig. 3A-E and S32B. We agree that it remains challenging to identify markers exclusive to AGM haematopoiesis, which is operationally equated with generation of transplantable haematopoietic stem cells. While HSC generation is a key event characteristic of the AGM, not all AGM haematopoiesis corresponds to HSCs, an important point in evaluating the data presented in the manuscript, and one that is acknowledged by us. The main text has been edited to clarify the experiments pertaining to distinguishing AGM and YS haematopoiesis, which are detailed in lines 180-187, 200-221, 268-304, and 315-356.

Following on the Reviewer’s comments about Cd34, we also inspected co-expression of Cd34 with Cd41 and Cd45, the latter co-expression present in, although not necessarily exclusive to, AGM haematopoiesis. Reassuringly, we observed clear co-expression with both markers (Author response image 1), in addition to a CD41+CD34- population, which likely reflects YS EMP-independent erythropoiesis. Flow cytometry analysis of co-expression of CD31 and CD34 in CD41+ and CD45+ populations at 144h and 216h timepoints has been included in Fig. 2B-D, Fig. 2S1A-D, including controls. In text: 180-187. We have earlier on in the rebuttal highlighted the fact that marker expression is responsive to the levels of Activin A used in the patterning pulse, with the 100ng/ml Activin A used in our protocol superior to 75ng/ml.

**Author response image 1. sa4fig1:** Association of CD34 with CD41 and CD45 expression is Activin A-responsive and supports the presence of definitive haematopoiesis. A. Flow cytometry analysis of CD34 and CD41 expression in 216h-haemogenic gastruloids; two doses of Activin A were used in the patterning pulse with CHI99021 between 48-72h. FMO controls shown. B. Flow cytometry analysis of CD34 and CD45 at 216h in the same experimental conditions.

Given the centrality of this point in comments by all the Reviewers, we have conducted projections of our single-cell RNA-seq data against two studies which (1) capture arterial and haemogenic specification in the para-splanchnopleura (pSP) and AGM region between E8.0 and E11 (Hou et al, PMID: 32203131), and (2) uniquely capture YS, AGM and FL progenitors and the AGM endothelial-tohaematopoietic transition (EHT) in the same scRNA-seq dataset (Zhu et al, PMID: 32392346). Focusing the analysis on the subsets of haemogenic gastruloid cells sorted as CD41+ (144h) C-Kit+ (144h and 192h) and CD45+ (192h and 216h) (now represented in Fig. 3A, and projected onto the studies in Fig. 4A), we show:

(1) That a subset of haemato-endothelial cells from haemogenic gastruloids at 144h to 216h project onto intra-embryonic cells spanning E8.25 to E10 (revised Fig. 4A left and 4S1A). This is in agreement with our original interpretation that 216h are no later than the MPP/pre-HSC state of embryonic development, requiring further maturation to generate engrafting progenitors. We have nevertheless removed specific references to pre-HSC, and instead referred to HSPC/progenitors.

(2) That haemogenic gastruloids contain YS-like (including EMP-like) and AGM-like haematopoietic cells (Fig. 4A centre and 4 S1B). Significantly, some of the cells, particularly CKit-sorted cells with a candidate endothelial and HE-like signature project onto AGM pre-HE and HE, as well as IAHC. Some 144h CD41+ and 192h CD45+ cells also project onto IAHC, suggesting that YS-like and AGM-like programmes arise independently and with partial timedependent organisation in the haemogenic gastruloid model. Later, predominantly 216h cells, have characteristics of MPP/LMPP-like cells from the FL, suggesting a progenitor wave of differentiation.

Altogether, the data support the notion that haemogenic gastruloids capture YS and AGM haematopoiesis until E10, as suggested by us in the manuscript.This re-analysis of the scRNA-seq data which was indeed prompted by challenging and insightful comments from the Reviewers, has been incorporated in the manuscript as described above and further listed here:

Re-clustering and highlights of specific markers in our scRNA-seq data in Figure 3A-E. In text: 268-304.

Projections to mouse embryo datasets in Figure 4A (Figure 4S1A-C; Supplementary File 3). In text: 315-356.

Single-cell RNA sequencing was used to compare hGx with mouse AGM. The authors incorrectly conclude that ' ..specification of endothelial and HE cells in hGx follows with time-dependent developmental progression into putative AGM-like HE..' And, '...HE-projected hGx cells.......expressed Gata2 but not Runx1, Myb, or Gfi1b..' Hemogenic endothelium is defined by the expression of Runx1 and Gfli1b is downstream of Runx1.

As a hierarchy of regulation, Gata2 precedes and drives Runx1 expression at the specification of HE (PMID: 17823307; PMID: 24297996), while Runx1 drives the EHT, upstream of Gfi1b in haematopoietic clusters (PMID: 34517413). Please note that the text segment the Reviewer refers to has been removed from the manuscript, as the analysis is no longer solely focused on projection to Thambyrajah et al (2024) data, and instead gained significantly from the projections on to the Hou et al (2020) and Zhu et al (2020) studies, as detailed above.

(3) The hGx protocol 'generates hematopoietic SC precursors capable of short-term engraftment' is not supported by the data presented. Short-term engraftment would be confirmed by flow cytometric detection of hematopoietic cells within the recipient bone marrow, spleen, thymus, and peripheral blood that expressed the BFP transgene. This analysis was not provided. PCR detection of transcripts, following an unspecified number of amplification cycles, as shown in Figure 3G (incorrectly referred to as Figure 3F in the legend) is not acceptable evidence for engraftment.

We provide the full flow cytometry analysis of spleen engraftment in the 5 mice which received implantation of 216h-haemogenic gastruloids in the adrenal gland and were analysed at 4 weeks; an additional (control) animal received adrenal injection of PBS (Fig. 4B-D in the revised manuscript). In this experiment, the bone marrow collection was limiting, and material was prioritised for PCR (Fig. 4C and full gels in 4S2C in the revised manuscript).

We had previously provided only representative plots of flow cytometry analysis of bone marrow and spleen, which we described as low-level engraftment and were chosen conservatively. The analysis was meant to complement the genomic DNA PCR, where detection was present in only some of the replicates tested per animal. On this note, we confirm that PCR analysis used conventional 40 cycles; the sensitivity had already been shown in the earlier version of the manuscript and is again represented in Fig. 4S2B. We argue that the low level of cytometric and molecular engraftment at 4 weeks, from haemogenic gastruloid-derived progenitors that have not progressed beyond a stage equivalent to E10 (Fig. 4A and Supplementary File 3 in the revised manuscript from scRNAseq projections), and that we have described as requiring additional maturation in vivo, are not surprising. Indeed, as previously shown and now repeated in in Fig. 2B-E (controls in Fig. 2S1E-G) in the revised manuscript, no more than 7 CD45+CD144+ multipotent cells are present per haemogenic gastruloid. We are only able to implant 3 haemogenic gastruloids in the adrenal gland of each transplanted animal.

We have rephrased Results and Discussion in lines 359-415 and 588-621, respectively, to rectify the nature of the engraftment, which we now attribute more generically to progenitors, also in light of the developmental time we could capture in the gastruloids prior to implantation.

Transplanted hGx formed teratoma-like structures, with hematopoietic cells present at the site of transplant only analysed histologically. Indeed, the quality of the images provided does not provide convincing validation that donor-derived hematopoietic cells were present in the grafts.

As stated in the text, the images mean to illustrate that the haemogenic gastruloids developed in situ. Further analysis motivated by the Reviewers’ comments and indeed a subsequent experiment with analysis of engraftment at a later timepoint of 8 weeks (revised Fig. 4E and 4 S2F-G) did not show a direct correspondence between engraftment and in vivo development or expansion, although this occurs in some cases. To be clearer, the observation of donor-derived blood cells in the implanted haemogenic gastruloids would not correspond to engraftment, as we have amply demonstrated that they have generated blood cells in vitro. There is no evidence that there are remaining pluripotent cells in the haemogenic gastruloid after 9 days of differentiation, and it is therefore not clear that the structures observed are teratomas. We specifically comment on this point in the revised manuscript – lines 601-607.

There is no justification for the authors' conclusion that '... the data suggest that 216h hGx generate AGM-like pre-HSC capable of at least short-term multilineage engraftment upon maturation...'. Indeed, this statement is in conflict with previous studies demonstrating that pre-HSCs in the dorsal aorta of the mouse embryo are immature and actually incapable of engraftment.

We have clearly stated that we do not see haematopoietic engraftment through transplantation of dissociated haemogenic gastruloids, which reach the E10 state containing pre-HSC (revised Fig 4A, 4S1A and Supplementary File 3). Instead, we observed rare myelo-erythroid (revised Fig. 4S2F-G) and myelo-lymphoid (revised Fig. 4E) engraftment upon in vivo maturation of haemogenic gastruloids with preserved 3D organisation. These statements are not contradictory. Nevertheless, we have now more cautiously attributed engraftment to the present of progenitors as a generic designation, and not to pre-HSC (lines 412-414 and 588-592 in the revised manuscript).

The statement '...low-level production of engrafting cells recapitulates their rarity in vivo, in agreement with the embryo-like qualities of the gastruloid system....' is incorrect. Firstly, no evidence has been provided to show the hGx has formed a dorsal aorta facsimile capable of generating cells with engrafting capacity. Secondly, although engrafting cells are rare in the AGM, approximately one per embryo, they are capable of robust and extensive engraftment upon transplantation.

As indicated above, the statement in lines 412-414 now reads “Engraftment is erythromyeloid at 4 weeks and lympho-myeloid at 8 weeks, reflecting different classes of progenitors, putatively of YS-like and AGM-like affiliation.” To be clear, with our original statement we meant to highlight that the production of definitive AGM-like haematopoietic progenitors (not all of which are engrafting) in haemogenic gastruloids does not correspond to non-physiological single-lineage programming. We did and do not claim that we achieved production of HSC, which would be long-term engrafting.

(4) Expression MNX1 transcript and protein in hematopoietic cells in MNX1 rearranged acute myeloid leukaemia (AML) is one cause of AML in infants. In the hGX model of this disease, Mnx1 is overexpressed in the mESCs that are used to form gastruloids. Mnx1 overexpression seems to confer an overall growth advantage on the hGx and increase the serial replating capacity of the small number of hematopoietic cells that are generated. The inefficiency with which the hGx model generates hematopoietic cells makes it difficult to model this disease. The poor quality of the cytospin images prevents accurate identification of cells. The statement that the kit-expressing cells represent leukemic blast cells is not sufficiently validated to support this conclusion. What other stem cell genes are expressed? Surface kit expression also marks mast cells, frequently seen in clonogenic assays of blood cells. Flow cytometric and gene expression analyses using known markers would be required.

The haemogenic gastruloid model generates haematopoietic and haemato-endothelial cells. MNX1 expands C-Kit+ cells at 144h, which we show to have a haemato-endothelial signature (see revised Fig. 3A-E, Supplementary File 2). We have added additional flow cytometry data showing that the replating cells from MNX1 express CD31 (Figure 6S1A-B).

Serial replating of CFC assays is a conventional in vitro assay of leukaemia transformation. Critically, colony replating is not maintained in EV control cells, attesting to the transformation potential of MNX1. Although we have not fully-traced the cellular hierarchy of MNX1-driven transformation in the haemogenic gastruloid system, the in vitro replating expands a C-Kit+ cell (revised Fig. 6E), which reflects the surface phenotype of the leukaemia, also recapitulated in the mouse model initiated by MNX1-overexpressing FL cells. Importantly, it recapitulates the transcriptional profile of MNX1leukaemia patients (revised Fig. 7C), which is uniquely expressed by MNX1144h and replated colony cells, but not to MNX1 216h gastruloid cells, arguing against a generic signature of MNX1 overexpression (revised Fig. 7B). Importantly, the MNX1-transformation of haemogenic gastruloid cells is superior to the FL leukaemia model at capturing the unique transcriptional features of MNX1-driven leukaemia, distinct from other forms of AML in the same age group (Fig 7 S1D-F). It is possible that this corresponds to a pre-leukaemia event, and we will explore this in future studies, which are beyond the proof-of-principle nature of this paper.

(5) In human infant MNX1 AML, the mutation is thought to arise at the fetal liver stage of development. There is no evidence that this developmental stage is mimicked in the hGx model.

We never claim that the haemogenic gastruloid model mimics the foetal liver. We propose that susceptibility to MNX1 is at the HE-to-EMP transition. Moreover, and importantly, contrary to the Reviewer’s statement, there is no evidence in the literature that the mutation arises in the foetal liver stage, just that the mutation arises before birth (PMID: 38806630), which is different. In a mouse model of MNX1 overexpression, the authors achieve leukaemia engraftment upon MNX1 overexpression in foetal liver, but not in bone marrow cells (PMID: 37317878). This is in agreement with a vulnerability of embryonic / foetal, but not adult cells to the MNX1 expression caused by the translocation. However, haematopoietic cells in the foetal liver originate from YS and AGM precursors, so the origin of the MNX1susceptible cells can be in those locations, rather than the foetal liver itself.

**Reviewer #2 (Public review):**
Summary:In this manuscript, the authors develop an exciting new hemogenic gastruloid (hGX) system, which they claim reproduces the sequential generation of various blood cell types. The key advantage of this cellular system would be its potential to more accurately recapitulate the spatiotemporal emergence of hematopoietic progenitors within their physiological niche compared to other available in vitro systems. The authors present a large set of data and also validate their new system in the context of investigating infant leukemia.Strengths:The development of this new in vitro system for generating hematopoietic cells is innovative and addresses a significant drawback of current in vitro models. The authors present a substantial dataset to characterize this system, and they also validate its application in the context of investigating infant leukemia.Weaknesses:The thorough characterization and full demonstration that the cells produced truly represent distinct waves of hematopoietic progenitors are incomplete. The data presented to support the generation of late yolk sac (YS) progenitors, such as lymphoid cells, and aortic-gonad-mesonephros (AGM)-like progenitors, including pre-hematopoietic stem cells (pre-HSCs), by this system are not entirely convincing. Given that this is likely the manuscript's most crucial claim, it warrants further scrutiny and direct experimental validation. Ideally, the identity of these progenitors should be further demonstrated by directly assessing their ability to differentiate into lymphoid cells or fully functional HSCs. Instead, the authors primarily rely on scRNA-seq data and a very limited set of markers (e.g., Ikzf1 and Mllt3) to infer the identity and functionality of these cells. Many of these markers are shared among various types of blood progenitors, and only a well-defined combination of markers could offer some assurance of the lymphoid and pre-HSC nature of these cells, although this would still be limited in the absence of functional assays.The identification of a pre-HSC-like CD45⁺CD41⁻/lo C-Kit⁺VE-Cadherin⁺ cell population is presented as evidence supporting the generation of pre-HSCs by this system, but this claim is questionable. This FACS profile may also be present in progenitors generated in the yolk sac such as early erythromyeloid progenitors (EMPs). It is only within the AGM context, and in conjunction with further functional assays demonstrating the ability of these cells to differentiate into HSCs and contribute to long-term repopulation, that this profile could be strongly associated with pre-HSCs. In the absence of such data, the cells exhibiting this profile in the current system cannot be conclusively identified as true pre-HSCs.

We present 2 additional pieces of evidence to support our claims that we capture YS and AGM stages of haematopoietic development.

(I) In the new Figures 4A and 4 S1A-C and Supplementary File 3 in the revised manuscript, we project our single-cell RNA-seq data onto (1) developing intra-embryonic pSP and AGM between E8 and E11 (Fig. 4A left, 4S1A) and (2) a single-cell RNA-seq study of HE development which combines haemogenic and haematopoietic cells from the YS, the developing HE and IAHC in the AGM, and FL (Fig. 4A centre, 4S1B). Our data maps E8.25-E10, and captures YS EMP and erythroid and myeloid progenitors, as well as AGM pre-HE, HE and IAHC, with some cells matching HSPC and LMPP, as suggested by the projection onto the Thambyrajah et al data set (already presented in the previous version of the manuscript, and now in Fig. 4A right and 4 S1C). The projection of the scRNA-seq data in presented in lines 314-355 of the revised manuscript. The scRNA-seq data itself was refocused on haemato-endothelial programmes as presented in the revised Fig. 3A-E, described in lines 267-303.

(II) Given the difficulty in finding markers that specifically associate with AGM haematopoiesis, we inspected the possibility of capturing different regulatory requirements at different stages of gastruloid development mirroring differential effects in the embryo. Polycomb EZH2 is specifically required for EMP differentiation in the YS, but does not affect AGM-derived haematopoiesis; it is also not required for primitive erythroid cells (PMID: 29555646; PMID: 34857757). We treated haemogenic gastruloids from 120h onwards with either DMSO (0.05%) or GSK126 (0.5uM), and inspected the cellularity of gastruloids at 144h, which we equate with YS-EMP, and 216h – putatively AGM haematopoiesis. We show that EZH2 inhibition / GSK126 treatment specifically reduces %CD41+ cells at 144h, but does not reduce %CD41+ or %CD45+ cells at 216h. We have included this experiment in the manuscript in Fig. 2 S2B-C (in text: 209-221).

These data, together with the scRNA-seq projections described, provide evidence to our claim that 144h haemogenic gastruloids capture YS EMPs, while CD41+ and CD45+ cells isolated at 216h reflect AGM progenitors. We cannot conclude as to the functional nature of the AGM cells from this experiment. The main text has been edited to clarify the experiments pertaining to distinguishing AGM and YS haematopoiesis (lines 180-187; 200-221; 268-304; 315-356).

The engraftment data presented are also not fully convincing, as the observed repopulation is very limited and evaluated only at 4 weeks post-transplantation. The cells detected after 4 weeks could represent the progeny of EMPs that have been shown to provide transient repopulation rather than true HSCs.

In the original version of the manuscript, we stated that there is low level engraftment and did not claim to have generated HSC. Instead, we described cells with short-term engraftment potential. We agree with the Reviewer that the cells we show in the manuscript at 4 weeks could be EMPs (revised Fig. 4B-E and 4 S2D-G). Additionally, we now have 8-week analysis of implant recipients, in which we observed, again low-level, a multi-lineage engraftment of the recipient bone marrow in 1:3 recipients (revised Fig. 4B-E and 4S2F-H). This engraftment is myeloid-lymphoid and therefore likely to have originated in a later progenitor. To be clear, we do not claim that this corresponds to the presence of HSC. It nevertheless supports the maturation of progenitors with engraftment potential. Limiting amounts of material was prioritised for flow cytometry stainings, not allowing PCR analysis. We rephrased Results and Discussion in lines 359-414 and 588-621, respectively, to rectify the nature of the engraftment.

**Reviewer #3 (Public review):**
In this study, the authors employ a mouse ES-derived "hemogenic gastruloid" model which they generated and which they claim to be able to deconvolute YS and AGM stages of blood production in vitro. This work could represent a valuable resource for the field. However, in general, I find the conclusions in this manuscript poorly supported by the data presented. Importantly, it isn't clear what exactly are the "YS" and the "AGM"-like stages identified in the culture and where is the data that backs up this claim. In my opinion, the data in this manuscript lack convincing evidence that can enable us to identify what kind of hematopoietic progenitor cells are generated in this system. Therefore, the statement that "our study has positioned the MNX1-OE target cell within the YS-EMP stage (line 540)" is not supported by the evidence presented in this study. Overall, the system seems to be very preliminary and requires further optimization before those claims can be made.Specific comments below:(1) The flow cytometric analysis of gastruloids presented in Figure 1 C-D is puzzling. There is a large % of C-Kit+ cells generated, but few VE-Cad+ Kit+ double positive cells. Similarly, there are many CD41+ cells, but very few CD45+ cells, which one would expect to appear toward the end of the differentiation process if blood cells are actually generated. It would be useful to present this analysis as consecutive gating (i.e. evaluating CD41 and CD45 within VE-Cad+ Kit+ cells, especially if the authors think that the presence of VE-Cad+ Kit+ cells is suggestive of EHT). The quantification presented in D is misleading as the scale of each graph is different.

Fig. 1C-D provide an overview of haemogenic markers during the timecourse of haemogenic gastruloid differentiation, and does indeed show a late up-regulation of CD45, as the Reviewer points out would be expected. The %CD45+ cells is indeed low. However, we should point out that the haemogenic gastruloid protocol, although biased towards mesodermal outputs, does not aim to achieve pure haematopoietic specification, but rather place it in its embryo-like context. We refute that the scale is misleading: it is a necessity to represent the data in a way that is interpretable by the reader: and we made sure from the outset that the gates (in C) are truly representative and annotated, as are the plot axes (in D). Consecutive gating at the 216h-timepoint is shown and quantified in Fig. 2S1D-F, or in the alternative consecutive gating suggested by the Reviewer, in Author response iamge 2 below. At the request of Reviewer 1, we also analysed CD31 and CD34 within CD41 and CD45 populations, again as validation of the emergent haematopoietic character of the cells obtained. This new analysis is shown in revised Fig. 2B, quantified in 2C.

**Author response image 2. sa4fig2:** Flow cytometry analysis of VE-cadherin+ cells in haemogenic gastruloids at 216h of the differentiation protocol, probing co-expression of CD45, CD41 and C-Kit.

(2) The imaging presented in Figure 1E is very unconvincing. C-Kit and CD45 signals appear as speckles and not as membrane/cell surfaces as they should. This experiment should be repeated and nuclear stain (i.e. DAPI) should be included.

We included the requested immunofluorescence staining in Figure 1E (216h). We also show the earlier timepoint of 192h here as Author response image 3. In text: lines 158-162.

**Author response image 3. sa4fig3:** Confocal images of haematopoietic production in haemogenic gastruloids. Wholemount, cleared haemogenic gastruloids were stained for CD45 (pseudo-coloured red) and C-Kit antigens (pseudo-coloured yellow) with indirect staining, as described in the manuscript. Flk1-GFP signal is shown in green. Nuclei are contrasted with DAPI. (A) 192h. (B) 216h.

(3) Overall, I am not convinced that hematopoietic cells are consistently generated in these organoids. The authors should sort hematopoietic cells and perform May-Grunwald Giemsa stainings as they did in Figure 6 to confirm the nature of the blood cells generated.

It is factual that the data are reproducible and complemented by functional assays shown in revised Fig. 2D-E, which clearly demonstrate haematopoietic output. The single-cell RNA-seq data also show expression of a haematopoietic programme, which we have complemented with biologically independent qRT-PCR analysis of the expression of key endothelial and haematopoietic marker and regulatory genes (revised Fig. 2F; in text: 200-209). As requested, we include Giemsa-Wright’s stained cytospins obtained at 216h to illustrate haematopoietic output. These are shown in revised Fig. 2S2A, in text: lines 194-199. Inevitably, the cytospins will be inconclusive as to the presence of endothelial-tohaematopoietic transition or the generation of haematopoietic stem/progenitor cells, as these cells do not have a distinctive morphology.

(4) The scRNAseq in Figure 2 is very difficult to interpret. Specific points related to this: - Cluster annotation in Figure 2a is missing and should be included.Why do the heatmaps show the expression of genes within sorted cells? Couldn't the authors show expression within clusters of hematopoietic cells as identified transcriptionally (which ones are they? See previous point)? Gene names are illegible.I see no expression of Hlf or Myb in CD45+ cells (Figure 2G). Hlf is not expressed by any of the populations examined (panels E, F, G). This suggests no MPP or pre-HSC are generated in the culture, contrary to what is stated in lines 242-245. (PMID 31076455 and 34589491).Later on, it is again stated that "hGx cells... lacked detection of HSC genes like Hlf, Gfi1, or Hoxa9" (lines 281-283). To me, this is proof of the absence of AGM-like hematopoiesis generated in those gastruloids.

For a combination of logistic and technical reasons, we performed single-cell RNA-seq using the Smart-Seq2 platform, which is inherently low throughput. We overcame the issue of cell coverage by complementing whole-gastruloid transcriptional profiling at successive time-points with sorting of subpopulations of cells based on individual markers documented in Fig. 1. We clearly stated which platform was used as well as the number and type of cells profiled (Fig. 3S1 and lines 226-241 of the revised manuscript), and our approach is standard. Following suggestions of the Reviewers to further focus our analysis on the haemogenic cellular differentiation within the gastruloids, we revised the presentation of the scRNA-seq data to now provide UMAP projections with representation and quantification of individual genes, including the ones queried by the Reviewer in Fig. 3 and respective supplements. Specifically, re-clustering and highlighting of specific markers are shown in Figure 3A-D and presented in lines 267-303 of the revised manuscript. Complementary independent real-time quantitative (q)PCR analysis showing time-dependent expression of endothelial and haematopoietic markers is now in Figure 2F. In text: 200-208.

(5) Mapping of scRNA-Seq data onto the dataset by Thambyrajah et al. is not proof of the generation of AGM HE. The dataset they are mapping to only contains AGM cells, therefore cells do not have the option to map onto something that is not AGM. The authors should try mapping to other publicly available datasets also including YS cells.

We have done this and the data are presented in Figure 4A (Figure 4S1A) and Supplementary File. In text: 314-355. As detailed in response to Reviewer 1, we have conducted projections of our single-cell RNA-seq data against two studies which (1) capture arterial and haemogenic specification in the para-splanchnopleura (pSP) and AGM region between E8.0 and E11 (Hou et al, PMID: 32203131) (revised Fig. 4A and 4 S1A), and (2) uniquely capture YS, AGM and FL progenitors and the AGM endothelial-to-haematopoietic transition (EHT) in the same scRNA-seq dataset (Zhu et al, PMID: 32392346) (revised Fig. 4A and 4 S1B). Specifically in answering the Reviewers’ point, we show that different subsets of haemogenic gastruloid cells sorted on haemogenic surface markers C-Kit, CD41 and CD45 cluster onto pre-HE and HE, intra-aortic clusters and FL progenitor compartments, and to YS EMP and erythroid and myeloid progenitors. This lends support to our claim that the haemogenic gastruloid system specifies both YS-like and AGM-like cells. Please note that we now do point out that some CD41+ cells at 144h project onto IAC, as do cells at the later timepoints, suggesting that AGM-like and YS-EMP-like waves may overlap at the 144h timepoint (lines…). In the future, we will address specific location of these cells, but that corresponds to a largescale spatial transcriptomics analysis requiring extensive optimisation for section capture which is beyond the scope of this manuscript and this revision.

(6) Conclusions in Figure 3, named "hGx specify cells with preHSC characteristics" are not supported by the data presented here. Again, I am not convinced that hematopoietic cells can be efficiently generated in this system, and certainly not HSCs or pre-HSCs.

We have provided evidence in the original manuscript, and now through additional experiments, that there is haematopoietic specification, including of progenitor cells, in the haemogenic gastruloid system. Molecular markers are shown in revised Fig. 2F and Fig. 3 and supplements; CFC assays are shown in revised Fig. 2D-E; cytospins are in revised Fig. 2 S2A; further analysis of 4-week implants and new analysis of 8-week implants (discussed below) are in revised Fig. 4 B-D and Fig. 4 S2 and we discussed the new scRNA-seq projections above. Importantly, we have never claimed, and again do not, that haemogenic gastruloids generate HSC. We accept the Reviewer’s comment that we have not provided sufficient evidence for the specification of pre-HSC-like cells and accordingly now refer more generically and conservatively to progenitors.

FACS analysis in 3A is again very unconvincing. I do not think the population identified as C-Kit+ CD144+ is real. Also, why not try gating the other way around, as commonly done (e.g. VE-Cad+ Kit+ and then CD41/CD45)?

Our gating strategy is not unconventional, which was done from a more populated gate onto the less abundant one to ensure that the results are numerically more robust. In the case of haemogenic gastruloids, unlike the AGM preparations the Reviewer may be referring to, CD41 and CD45+ cells are more abundant as there is no circulation of more differentiated haematopoietic cells away from the endothelial structures. This said, we did perform the gating as suggested (Rev Fig. 2), indeed confirming that most VE-cad+ Kit+ cells are CD45+. Interestingly VE-cad+Kit- are predominantly CD41+, reinforcing the haematopoietic nature of these cells.

The authors must have tried really hard, but the lack of short- or long-engraftment in a number of immunodeficient mouse models (lines 305-313) really suggests that no blood progenitors are generated in their system. I am not familiar with the adrenal gland transplant system, but it seems like a very non-physiological system for trying to assess the maturation of putative pre-HSCs. The data supporting the engraftment of these mice, essentially seen only by PCR and in some cases with a very low threshold for detection, are very weak, and again unconvincing. It is stated that "BFP engraftment of the Spl and BM by flow cytometry was very low level albeit consistently above control (Fig. S4E)" (lines 337-338). I do not think that two dots in a dot plot can be presented as evidence of engraftment.

We have presented the data with full disclosure and do not deny that the engraftment achieved is low-level and short-term, indicating incomplete maturation of definitive haematopoietic progenitors in the current haemogenic gastruloid system. Indeed, by not wanting to overstate the finding, we were deliberately conservative in our representative flow cytometry plots and focused on the PCR for sensitivity. We now present the full flow cytometry analysis for spleen where we preserved more cells after the genomic DNA extraction (revised Fig. 4C) and call the Reviewer’s attention to the fact that detection of BFP+ cells by PCR and flow cytometry in the recipient animals is consistent between the 2 methods (revised Fig. 4C and D; full gels previously presented now in Fig. 4S2C; sensitivity analysis was also previously available and is now in Fig. 4S2B). In addition, we have now also been able to detect low-level myelo-lymphoid engraftment in the bone marrow and spleen 8 weeks after adrenal implantation, again suggesting the presence of a small number of definitive haematopoietic progenitors that potentially mature from the 3 haemogenic gastruloids implanted (Fig. 4E and 4 S2F-G in the revised manuscript). We rephrased Results and Discussion at lines 359-414 and 589-621, respectively, to rectify the nature of the engraftment which we attribute to progenitors.

(7) Given the above, I find that the foundations needed for extracting meaningful data from the system when perturbed are very shaky at best. Nevertheless, the authors proceed to overexpress MNX1 by LV transduction, a system previously shown to transform fetal liver cells, mimicking the effect of the t(7;12) AML-associated translocation. Comments on this section:The increase in the size of the organoid when MNX1 is expressed is a very unspecific finding and not necessarily an indication of any hematopoietic effect of MNX1 OE.

We agree with the Reviewer on this point; it is nevertheless a reproducible observation which we thought relevant to describe for completeness and data reproducibility.

The mild increase of cKit+ cells (Figure 4E) at the 144hr timepoint and the lack of any changes in CD41+ or CD45+ cells suggests that the increase in Kit+ cells % is not due to any hematopoietic effect of MNX1 OE. No hematopoietic GO categories are seen in RNA seq analysis, which supports this interpretation. Could it be that just endothelial cells are being generated?

The Reviewer is correct that the MNX1-overexpressing cells have a strong endothelial signature, which is present in patients (revised Fig. 5A). We investigated a potential link with C-Kit by staining cells from the replating colonies during the process of in vitro transformation with CD31. We observed that 40-50% of C-Kit+ cells (20-30% total colony cells) co-expressed CD31, at least at early plating. These cells co-exist with haematopoietic cells, namely Ter119+ cells, as expected from the YSlike erythroid and EMP-like affiliation of haematopoietic output from 144h-haemogenic gastruloids. These data are included in Fig. 6S1A-B (in text 506-507) of the revised manuscript.

(8) There seems to be a relatively convincing increase in replating potential upon MNX1-OE, but this experiment has been poorly characterized. What type of colonies are generated? What exactly is the "proportion of colony forming cells" in Figures 5B-D? The colony increase is accompanied by an increase in Kit+ cells; however, the flow cytometry analysis has not been quantified.

Given the inability to replate control EV cells, there is not a population to compare with in terms of quantification. The level of C-Kit+ represented in Fig. 6E of the revised manuscript is achieved at plate 2 or 3 (depending on the experiment), both of which are significantly enriched for colony-forming cells relative to control (revised Fig. 6B, D).

(9) Do hGx cells engraft upon MNX1-OE? This experiment, which appears not to have been performed, is essential to conclude that leukemic transformation has occurred.

For the purpose of this study, we are satisfied with confirmation of in vitro transformation potential of MNX1 haemogenic gastruloids, which can be used for screening purposes. Although interesting, in vivo leukaemia engraftment from haemogenic gastruloids is beyond the scope of this study.

**Reviewer #2 (Recommendations for the authors):**
(1) Minor comments(a) I find the denomination "hGx" very confusing as it would suggest that these gastruloids are human, whereas, in fact, they are murine.

We agree with the Reviewer on the confusing nomenclature and have edited the manuscript to call “haemGx” instead.

(b) I find the presence of mast cells in CFC of MNX1-OE cultures very puzzling as this does not bear any resemblance to human leukemia.

We detect an enrichment of mast cell transcriptional programmes, as defined by the cell type repositories. While it is not mast cells to represent leukaemic cells in patients, this ontology is likely to reflect the developmental stage and origin of progenitors which are affected by MNX1.

(2) I have a few suggestions to improve figures and tables clarity, to help readers better follow the data presented.(a) To enhance readability, it would be beneficial to highlight the genes mentioned in the text within the scRNA-seq figures. Many figures currently display over 30-40 genes in small font sizes, making it difficult to quickly locate specific genes discussed in the text. Additionally, implementing a colorcoding system to categorize these genes according to their proposed lineages would improve clarity and organization.

We have now performed major re-organisation and re-analyses of the scRNA-seq data, which we believe has improved the readability and clarity of the corresponding sections of the manuscript.

(b) The data presented in Supplementary Table 1, along with other supplementary tables, are challenging to interpret due to insufficient annotations. Enhancing these tables with clearer and more detailed annotations would significantly improve clarity and aid readers in understanding the supplementary materials.

Descriptive text has been added to accompany each Supplementary File to aid in understanding the results reported therein.

**Reviewer #3 (Recommendations for the authors):**
In addition to what was written in the public review, I would suggest the authors simplify and shorten the text. Currently, a lot of unnecessary detail is included which makes the story very hard to follow. Moreover, the authors should modify the figures to make them more comprehensible, especially for RNA-seq data.

We have significantly re-arranged and shortened parts of the manuscript, particularly by focusing the Discussion. Results presentation has also been improved through additional analysis and graphic representation of the scRNA-seq data, which we believe has improved the readability and clarity.s